# Genetic Insights from Line × Tester analysis of Maize Lethal Necrosis testcrosses for developing multi-stress-resilient hybrids in Sub-Saharan Africa

**Manje Gowda**[1]*, **Yoseph Beyene**[1]*, **Suresh Lingadahalli Mahabaleswara**[1], **Veronica Ogugo**[1,2], **Manigben Kulai Amadu**[1,2,3], **Vijay Chaikam**[1]

**1** International Maize and Wheat Improvement Center (CIMMYT), World Agroforestry Centre (ICRAF), United Nations Avenue, Gigiri, Nairobi, Kenya, **2** West Africa Centre for Crop Improvement (CIMMYT), University of Ghana, Accra, Ghana, **3** CSIR-Savanna Agricultural Research Institute, Tamale, Nyankpala, Ghana

* m.gowda@cgiar.org (MG); y.beyene@cgiar.org (YB)

## Abstract

A systematic evaluation of maize hybrid performance and combining ability was conducted to enhance resistance to maize lethal necrosis (MLN), drought tolerance, and grain yield (GY) in eastern and southern Africa. Thirty-eight early- to intermediate-maturing maize inbred lines, including MLN-tolerant and high-yielding genotypes with drought tolerance and resistance to multiple foliar and insect pests, were crossed with 29 single-cross testers to generate 437 testcross hybrids. These hybrids were evaluated under managed MLN inoculation, drought stress, and optimum conditions across multiple locations. Continuous variation in GY, disease severity, and agronomic traits confirmed quantitative inheritance, with strong positive correlations between GY and ears per plant and negative correlations between MLN severity and yield. Variance analyses revealed highly significant genotypic and genotype × environment interactions, with additive effects predominating across environments (Baker's ratios 0.85–0.99; heritability 0.69–0.88), supporting effective selection based on general combining ability (GCA). Superior MLN-tolerant hybrids, such as (CKLMARSI0037/CKLTI0139)//CKDHL120312, achieved up to 5.75 t ha$^{-1}$ under MLN, exceeding commercial checks by over fivefold. Under optimum and drought conditions, top hybrids maintained high yield, foliar disease resistance, short anthesis–silking intervals, and delayed senescence. Specific combining ability (SCA) effects highlighted stress-specific non-additive interactions, particularly under drought, underscoring the need for targeted parental selection. GCA analyses identified across environment and environment-specific favorable parents, including CKDHL120312, CKDHL140910, CKLMARSI0037/CKLTI0139, and CML322/CML543, while GGE biplots confirmed tester discrimination and representativeness. These findings demonstrate that integrating MLN resistance, drought tolerance, and high

**Data availability statement:** All relevant data are within the manuscript and its Supporting Information files.

**Funding:** The research was supported by the Bill and Melinda Gates Foundation (B&MGF), and the United States Agency for International Development (USAID) through the Stress Tolerant Maize for Africa (STMA, B&MGF Grant # OPP1134248) Project, AGGMW (Accelerating Genetic Gains in Maize and Wheat for Improved Livelihoods, B&MGF Investment ID INV-003439) project and Resilient Maize Hybrids for Sub-Saharan Africa (GF Investment ID INV-088326). The funders had no role in this study design, data collection and analysis, decision to publish, or preparation of the manuscript.

**Competing interests:** The authors declare they have no conflict of interest.

yield is achievable without compromising other agronomic performance. The study provides a robust framework for selecting elite parents and testers, exploiting additive and non-additive genetic effects, and developing resilient, high-performing maize hybrids for sub-Saharan Africa.

## Introduction

Maize (*Zea mays* L.) is the most widely cultivated and consumed cereal crop in sub-Saharan Africa (SSA), grown on over 40 million hectares and providing more than 30% of daily caloric intake [1]. It is a foundation of food security, income generation, and rural livelihoods. However, average yields remain below 2 t ha$^{-1}$ mainly due to recurring drought and biotic stresses [2]. In eastern Africa, Maize Lethal Necrosis (MLN), caused by co-infection with *Maize chlorotic mottle virus* (MCMV) and *Sugarcane mosaic virus* (SCMV), has emerged as a major production constraint. Yield losses from MLN can be devastating, emphasizing the need for tolerant hybrids that maintain productivity under both optimal growing conditions and stress conditions such as drought, which is common across SSA.

The extent of drought-induced yield loss in maize can range up to 90%, depending on the stress intensity, timing, and genotype tolerance [1,3]. As a result, the development of drought-tolerant maize varieties has become a core breeding objective in SSA [4]. To address this, the International Maize and Wheat Improvement Center (CIMMYT), in partnership with National Agricultural Research Systems (NARS), has released a wide range of multiple stress tolerant maize hybrids, including drought-tolerance and adapted to diverse agroecologies. These efforts have delivered measurable genetic gains for grain yield under managed (32.5 kg ha$^{-1}$ year$^{-1}$) and random drought (22.7 kg ha$^{-1}$ year$^{-1}$) environments [1,5–7]. However, yield improvements remain insufficient to meet growing food demand driven by population increase, worsening climate variability, and limited input access in SSA [5,7,8]. Continued genetic improvement for multi-stress-tolerant, high-yielding hybrids remains a critical priority.

Developing stress-resilient hybrids requires genetically diverse parental lines and accurate estimates of their ability to transmit favorable alleles to offspring [9]. Combining ability analyses provide insights into gene action governing complex quantitative traits such as grain yield, flowering behavior, and stress adaptation. Line × tester method is efficient in maize breeding, allowing rapid evaluation of many new inbred lines against a few elite testers [10]. This facilitates the identification of lines contributing predominantly additive effects (general combining ability, GCA) or non-additive effects (specific combining ability, SCA), aiding in the selection of superior parental combinations. Early-stage test-cross evaluations in line-by-tester design helps to identify promising combiners, optimizes the resource use, and shortens the breeding cycles. By analyzing GCA and SCA, breeders can distinguish between lines with broad adaptability and those exhibiting hybrid-specific performance, which is critical in SSA's heterogeneous and stress-prone environments.

To represent the diversity of growing conditions, breeding programs employ multi-environment trials (METs) across various agroecological zones and management levels [11]. However, because natural stress occurrence is often unpredictable, CIMMYT and partners have established managed stress screening sites that simulate drought, low-nitrogen, and MLN conditions in a controlled and repeatable manner. Over the past three decades, CIMMYT and NARS partners have achieved notable success in developing maize germplasm with improved tolerance to drought, and MLN [12–16]. Improved inbred lines from these efforts have led to the release of multiple-stress-resilient hybrids that combine yield stability with disease resistance and nutrient-use efficiency.

Several studies revealed varying contributions of additive and non-additive gene effects in controlling grain yield across environments [17–21]. Some studies observed that additive gene effects predominantly govern grain yield and flowering traits under drought, suggesting that recurrent selection could effectively improve tolerance [2,22]. Conversely, a stronger role for non-additive effects was also reported [23], emphasizing the importance of hybrid-specific performance under stress. These variations reflect differences in genetic backgrounds and the strong genotype × environment interactions typical of SSA production systems. Therefore, evaluating combining ability under both managed stress and optimal conditions remains essential for identifying stable and high-performing parental combinations.

Identifying suitable testers that can accurately differentiate maize inbred lines for combining ability, grain yield, MLN resistance, and drought tolerance is critical in maize breeding for SSA. It enables efficient use of genetic diversity, accelerates hybrid development, and enhances genetic gain. While notable advances have been achieved in developing drought- and disease-tolerant germplasm, the adoption of doubled haploid (DH) technology facilitates the rapid development of numerous fixed lines, necessitating continuous evaluation of newly developed elite DH lines. [16,24–26]. Assessing their combining ability across MLN, drought, and optimal environments provides valuable insights into parental performance and hybrid adaptability. Therefore, this study aimed to evaluate the genetic performance of recently developed tropical maize inbred lines and their testcross hybrids under MLN, managed drought, and optimal conditions using a line × tester mating design. The specific objectives were to: (i) assess grain yield and agronomic performance of testcross hybrids across stress and non-stress environments; (ii) estimate GCA and SCA effects for grain yield and related traits; (iii) identify superior hybrids that surpass commercial checks; and (iv) select testers capable of effectively classifying inbred lines for MLN resistance and overall performance. The findings will provide critical insights into the genetic potential of new inbred lines and support the development of high-yielding, multi-stress-resilient maize hybrids for smallholder production systems in sub-Saharan Africa.

## Materials and methods

### Germplasm

Thirty-eight early- to intermediate-maturing maize inbred lines developed by the International Maize and Wheat Improvement Center (CIMMYT), representing MLN-tolerant germplasm, lines exhibiting high general combining ability for grain yield and drought tolerance, resistance to multiple foliar and insect pests, and temperate introgression, were used in this study (Table 1). Twenty-nine single-cross testers (Table 1) were crossed to 38 maize elite inbred lines from a complementary heterotic group to form a total of 437 testcross hybrids.

### Experimental design

A total of 437 testcross hybrids plus seven commercial hybrids as checks (DK8031, WH505, Pioneer3253, Pioneer30G19, H517 and Duma43) and three MLN tolerant internal genetic checks (CKMLN150073, CKMLN150077, and CKMLN150079) were evaluated in multiple locations in Kenya. These selected locations are managed by CIMMYT and KALRO (Kenya Agricultural and Livestock Research Organization), working together on developing multiple stress-resilient hybrids for the region. The experiments were under artificial inoculation of MLN viruses, managed drought stress condition and optimal conditions using α-lattice with two replicates. The MLN infestation experiments were carried out in

**Table 1. List of inbred lines and single cross testers used in this study and their heterotic group (HG) information, and their special attributes.**

| Genotype | Name | HG | Comments |
|---|---|---|---|
| Line | CKDHL120312 | A | MLN tolerant, MSV tolerant line |
| Line | CKDHL120918 | A | MLN tolerant line |
| Line | CKDHL140475 | A | High GY, drought tolerant |
| Line | CKDHL140700 | A | High GY, drought tolerant |
| Line | CKDHL140910 | A | High GY, drought tolerant |
| Line | CKDHL141105 | A | High GY, drought tolerant |
| Line | CKDHL142425 | A | High GY, drought tolerant |
| Line | CKDHL142445 | A | High GY, drought tolerant |
| Line | CKDHL142806 | A | High GY, drought tolerant |
| Line | CKLTI0026 | A | ex-PVP temperate introgressed, tropical line |
| Line | CKLTI0043 | A | ex-PVP temperate introgressed, tropical line |
| Line | CKSBL10194 | A | Multiple insect tolerant |
| Line | CKSBL10205 | A | Multiple insect tolerant |
| Line | CML494 | B | MLN tolerant, intermediate maturing |
| Line | CML495 | A | Late maturing, good GCA for GY |
| Line | CKDHL120341 | B | High GY, drought tolerant |
| Line | CKDHL120358 | B | High GY, drought tolerant |
| Line | CKDHL120668 | B | High GY, drought tolerant |
| Line | CKDHL120694 | B | High GY, drought tolerant |
| Line | CKDHL140539 | B | High GY, drought tolerant |
| Line | CKDHL140548 | B | High GY, drought tolerant |
| Line | CKDHL143607 | B | High GY, drought tolerant |
| Line | CKLMARSI0022 | B | developed from MARS, drought tolerant, high GY |
| Line | CKLMARSI0029 | B | developed from MARS, drought tolerant, high GY |
| Line | CKLMLN150340 | B | MLN tolerant |
| Line | CKLMLN150356 | B | MLN tolerant |
| Line | CKLMLN150459 | B | MLN tolerant |
| Line | CKLMLN150461 | B | MLN tolerant |
| Line | CKLMLN150474 | B | MLN tolerant |
| Line | CKLMLN150478 | B | MLN tolerant |
| Line | CKLTI0133 | B | ex-PVP temperate introgressed tropical line |
| Line | CKLTI0134 | B | ex-PVP temperate introgressed tropical line |
| Line | CKLTI0136 | B | ex-PVP temperate introgressed tropical line |
| Line | CKLTI0139 | B | ex-PVP temperate introgressed tropical line |
| Line | CKLTI0230 | B | ex-PVP temperate introgressed tropical line |
| Line | CKLTI0318 | B | ex-PVP temperate introgressed tropical line |
| Line | CKMLN150478 | B | MLN tolerant line |
| Line | CML550 | B | MLN tolerant, low N tolerant |
| Tester | CKDHL0221/CKDHL120312 | A | MLN tolerant |
| Tester | CKDHL0221/CML464 | A | Multiple disease resistant |
| Tester | CKDHL120312/CKLTI0042 | A | MLN tolerant, high yielding |
| Tester | CKDHL120312/CML312 | A | MLN tolerant, drought tolerant |
| Tester | CKDHL120312/CML536 | A | MLN tolerant, high yielding |
| Tester | CKDHL120918/CML494 | A | MLN tolerant |
| Tester | CKLTI0043/CKDHL120312 | A | MLN tolerant, high yielding |
| Tester | CKSBL10060/CKDHL120312 | A | MLN tolerant, multiple insect resistance |

*(Continued)*

**Table 1.** (Continued)

| Genotype | Name | HG | Comments |
|---|---|---|---|
| Tester | CKSBL10194/CKDHL120312 | A | MLN tolerant, multiple insect resistance |
| Tester | CKSBL10205/CKDHL120312 | A | MLN tolerant, multiple insect resistance |
| Tester | CKDHL120918/CKLMARSI0022 | A/B | MLN tolerant, high GY, drought tolarant |
| Tester | CKDHL120918/CKLMARSI0029 | A/B | MLN tolerant, high GY |
| Tester | CKDHL120918/CKLTI0136 | A/B | MLN tolerant, high GY |
| Tester | CKDHL120918/CKLTI0138 | A/B | MLN tolerant, high GY |
| Tester | CKLMARSI0037/CKLTI0139 | B | Drought tolerant, high GY] |
| Tester | CKLMARSI0037/CML543 | B | Drought tolerant, high GY, foliar disease resistant |
| Tester | CKLTI0137/CKLMARSI0022 | B | MLN tolerant |
| Tester | CKLTI0137/CKLTI0330 | B | ex-PVP temperate introgressed tropical tester |
| Tester | CKLTI0138/CKLMARSI0022 | B | MLN tolerant, drought tolerant |
| Tester | CKLTI0138/CKLTI0330 | B | ex-PVP temperate introgressed tropical tester |
| Tester | CKLTI0139/CKLMARSI0022 | B | MLN tolerant, drought tolerant |
| Tester | CKLTI0139/CKLTI0335 | B | ex-PVP temperate introgressed tropical tester |
| Tester | CKLTI0227/CKLMARSI0022 | B | MLN tolerant, drought tolerant |
| Tester | CKLTI0227/CKLMARSI0029 | B | drought tolerant, high GY |
| Tester | CML322/CML543 | B | multiple foliar disease resistance |
| Tester | CML543/CML494 | B | multiple disease resistance |
| Tester | CKLTI0133/CKDHL120312 | B/A | MLN tolerant, high GY |
| Tester | CKLTI0139/CKDHL120918 | B/A | MLN tolerant, high GY |
| Tester | CKLTI0227/CKDHL120918 | B/A | MLN tolerant, high GY |

Ex-PVP lines: Inbreds with expired plant variety protection act certificates

two locations at Naivasha (34°45' E, 0°16' N, 1585masl) in Kenya and Babati (35°74' E, 4°21' N, 2145masl) in Tanzania. The optimum experiment were conducted in Kenya at Kakamega (34°45' E, 0°16' N, 1585masl), Kiboko (37°75' E, 2°15' S, 975masl), and Mbeere (37°43′E, 0°09′S, 1126 masl) and managed drought trials were evaluated at Kiboko (37°75' E, 2°15' S, 975masl). Optimal trials were planted between April and October, whereas managed drought trials were conducted between May and October. At Kiboko, Kakamega and Mbeere, optimal trials were planted earlier, while managed drought trials plantings were delayed to ensure a dry, rain-free period from flowering to harvest. All optimal and MLN trials were grown under irrigated or rainfed conditions, depending on location. For managed drought trials, irrigation was withdrawn two weeks prior to the expected flowering date to impose water stress during the flowering and grain-filling stages. Standard agronomic practices included the application of fertilizer at a rate of 60 kg N ha$^{-1}$ and 60 kg P$_2$O$_5$ ha$^{-1}$ as a basal fertilizer two weeks after planting, with a subsequent top-dressing of nitrogen (urea) at 60 kg N ha$^{-1}$ four weeks after planting. All experimental fields were maintained free of weeds.

## Screening for resistance to MLN

Testcross hybrids were artificially screened for resistance to MLN following the optimized protocol described in earlier studies [27,28] and in the MLN information portal (http://mln.cimmyt.org/mln-scoring). In brief, pure isolates of MCMV and SCMV were separately propagated and maintained in isolated screen houses using a susceptible maize genotype. The purity of each isolate was confirmed through ELISA testing. To prepare the inoculum, leaf samples were randomly collected from both SCMV and MCMV production screen houses, ground separately, and then mixed at an optimized ratio

of 4:1 (SCMV: MCMV). Inoculations were carried out twice—at the fifth and sixth weeks after planting—using a motorized backpack mist blower operating at a pressure of 10 kg cm⁻². The second inoculation was performed to minimize escapes within testing plots. MLN symptom severity was assessed two weeks after the final inoculation using a 1–9 rating scale (1 = no visible symptoms; 9 = complete plant necrosis). MLN susceptibility indices were computed from four consecutive ratings taken at 14-day intervals, starting from the first scoring date.

## Screening for GLS and TLB resistance

The testcross hybrids were evaluated for gray leaf spot (GLS) and turcicum leaf blight (TLB) under natural disease pressure at the Kakamega hotspot in Kenya. GLS severity, which typically peaks between tasseling and physiological maturity, was assessed at mid-silking and hard dough stages, while TLB severity was recorded at the hard dough stage. For both diseases, severity was scored plot-wise using a standardized ordinal scale ranging from 1 (highly resistant, no visible symptoms) to 9 (highly susceptible, extensive necrosis or plant death). For GLS, scores reflected increasing lesion number, size, and leaf area affected, ranging from clean plants or a few scattered lesions with <5% leaf damage (score 1) to abundant lesions on nearly all leaves, premature drying, or plant death with 86–100% leaf area affected (score 9). Intermediate scores captured progressive disease development, including moderate lesion abundance, chlorotic streaking, necrosis, and increasing leaf area damage. Similarly, TLB severity scores represented a continuum from no or very slight infection (score 1) to severe infection characterized by abundant lesions on most leaves, premature drying, or plant death with up to complete leaf area damage (score 9). Intermediate classes reflected increasing lesion abundance and vertical disease progression from lower to middle and upper canopy leaves, accompanied by a corresponding increase in the proportion of leaf area affected.

## Assessment of key agronomic traits

At each of the test sites, key agronomic traits were recorded. At flowering, days to anthesis (AD) was counted as the number of days from planting to when 50% of plants in a plot that shed pollen. Anthesis-silking interval (ASI) was calculated as the difference of days to silking (SD) and AD. Plant height (PH) in cm was measured as the average height of five randomly selected plants in a plot measured from the base of the plant to the first tassel branch. Ear height (EH) was taken as the average height of the same plants, measured from the base of the plant to the node bearing the uppermost ear. At harvest, grain yield (GY) in t ha⁻¹ was estimated from total plot weight adjusted to 12.5% moisture level. At harvest, number of ears per plant (EPP) was obtained by dividing the total number of ears per plot by the total number of plants harvested. Leaf senescence was visually assessed two weeks after flowering using a 1–10 numerical scale, where each score represents a 10% increment of dead total leaf area (1 = 10% and 10 = 100% dead leaf area).

## Statistical Analysis

Individual and combined environment analysis were performed according to the restricted maximum likelihood procedure using the multi-environment trial analysis program in R (META-R) [29] based on the following linear mixed:

$$y_{ijkl} = \mu + Env_i + Rep_j(Env_i) + Block_k\left(Env_iRep_j\right) + Gen_l + Env_i \times Gen_l + \varepsilon_{ijkl}$$

where $y_{ijkl}$ is the trait of interest; $\mu$ is the mean effect; $Env_i$ is the effects of the $i^{th}$ environment; $Rep_j(Env_i)$ is the effect of $i^{th}$ replicate nested within $i^{th}$ environment, $Block_k\left(Env_iRep_j\right)$ is the effect $k^{th}$ of the incomplete block within the $j^{th}$ replicate in the $i^{th}$ environment, $Gen_l$ is the effects of the $l^{th}$ genotype, $Env_i \times Gen_l$ is the genotype × environment interaction effect; and $\varepsilon_{ijkl}$ is the error associated with the $i^{th}$ environment; $j^{th}$ replicate, $k^{th}$ of the incomplete block, and $l^{th}$ genotype, which is assumed to be normally and independently distributed, with mean zero and homoscedastic variance. In this model, genotypes were considered fixed effects, calculating best linear unbiased estimator (BLUE),

whereas replications, blocks within replications, and environments (location and season combination) were considered random effects. To estimate variance components, all factors were considered random effects. Correlations between traits evaluated under MLN, managed drought and optimum conditions were computed using the cor function in R (R Core Team, 2024). Heatmaps depicting trait correlations, along with distributions of phenotypic values, were generated using the *ggplot2* package.

### Estimation of combining ability and variance components

The testcross data from optimum, MLN and drought management conditions were analyzed using the Analysis of Genetic Designs with R (AGD-R) software, Version 5.0 [30]. Both balanced and unbalanced L×T datasets were analyzed using the restricted maximum likelihood (REML) method for multi-environment lattice designs. This analysis provided estimates for the GCA of lines, GCA of testers, and SCA of L×T crosses. For each effect, the standard error, t-value, and probability were calculated for across locations within each management.

The analysis also included the estimation of variance components for lines, testers, L×T interactions, genotypes, additive effects, dominance effects, and environmental factors. Using these, both broad-sense and narrow-sense heritability were calculated [30]. In addition, Henderson's L×T multi-environment lattice method was used to perform the analysis of variance (ANOVA) across sites and to estimate GCA effects of lines for different traits.

To study the heterotic patterns between lines and testers, two-way L×T matrices were created and visualized using GGE biplot analysis. The biplots were generated using the "GGEModel" and/or "gge" functions from the GGEBiplots and metan packages in R [31,32]. These tables were prepared for across locations under MLN, optimum, and managed drought conditions using the adjusted mean values of GY. In the GGE biplot analysis, the grand mean of adjusted L×T GY data served as the reference point. Tester-centered (G+GE) biplots were then generated using singular-value decomposition (SVD) with the tester-focused (column metric preserving) method. This approach allowed visualization of the overall relationships between lines and testers and the heterotic grouping patterns among them.

## Results

A total of 38 inbred lines and 29 single-cross testers were used in this study, representing two major heterotic groups (HG A and HG B) (Table 1). Lines were diverse in origin and included MLN-tolerant, drought-tolerant, and insect-resistant genotypes derived from CIMMYT's doubled haploid and MARS breeding pipelines. Lines such as CKDHL120312, CKDHL120918, and CML494 were MLN- and maize streak virus (MSV)-tolerant. While several lines (e.g., CKDHL140475, CKDHL140700, CKDHL142806) were high-yielding and drought tolerant.

Testers included elite single crosses combining MLN tolerance and complementary heterotic backgrounds, such as CKDHL120918/CKLTI0138, CKLMARSI0037/CKLTI0139, and CML322/CML543, previously identified for yield stability and resistance to multiple diseases.

This diverse germplasm base provided a wide range of genetic variation suitable for estimating combining abilities and identifying promising hybrid combinations across multiple stress environments.

### Phenotypic correlations and trait relationships

Frequency distribution of 437 hybrids across locations under MLN (Supplementary Table S1 in S1 File), optimum (Supplementary Table S2 in S1 File), and drought conditions (Fig 1, Supplementary Table S3 in S1 File) showed continuous variation for GY, disease severity and agronomic traits, confirming their quantitative inheritance. Correlation analysis (Fig 2) revealed strong positive correlations between GY and EPP ($r > 0.60$), and negative correlations between MLN disease severity and GY ($r = -0.48$ to $-0.63$, $p < 0.01$). Anthesis and silking dates were strongly correlated ($r > 0.80$), whereas ASI exhibited a weak negative correlation with GY under drought, confirming its utility as a stress-adaptation indicator.

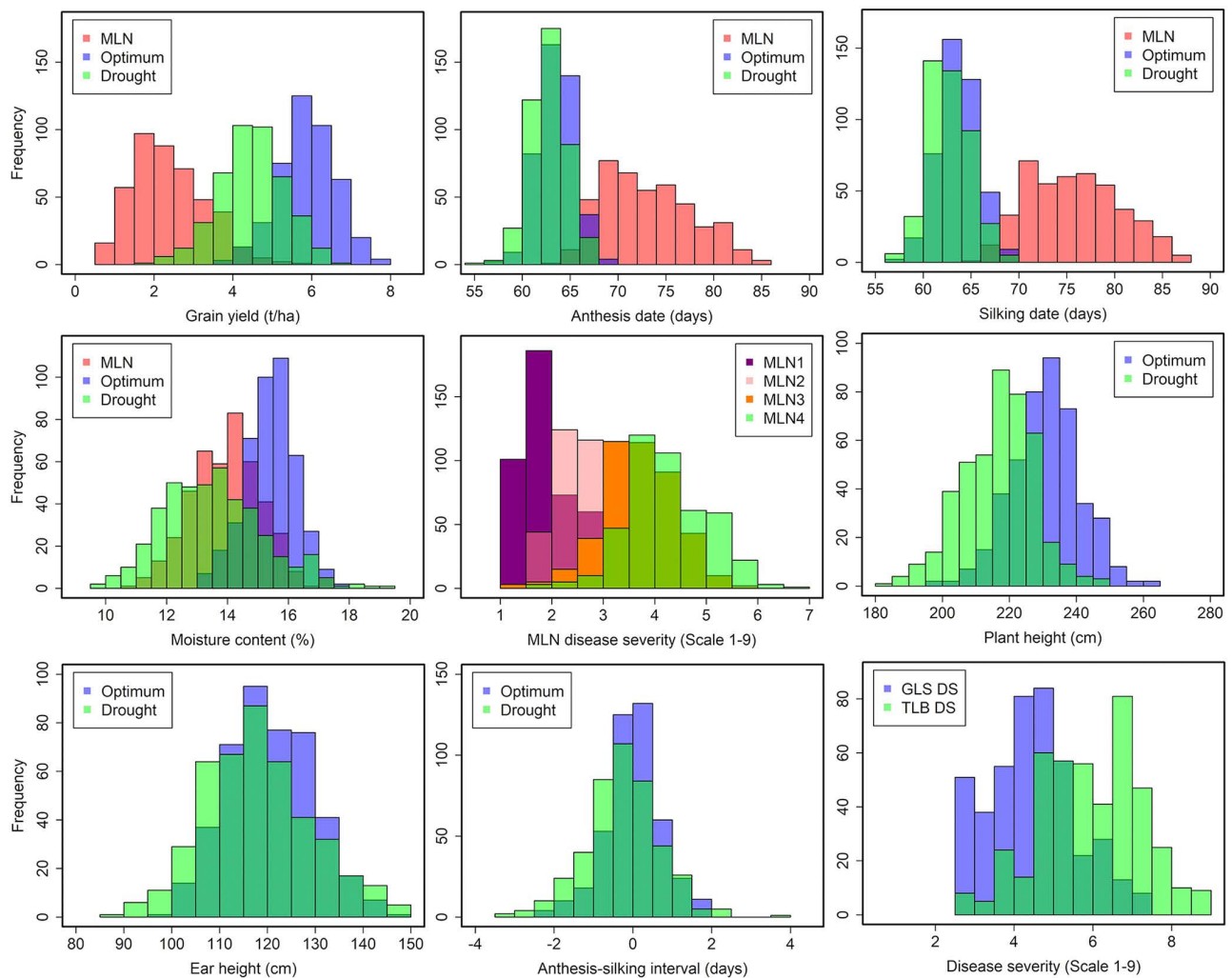

**Fig 1. Frequency distribution of 437 testcross hybrids for GY and other agronomic traits evaluated in under MLN disease pressure, optimum and drought management at different locations in Kenya.**

## Variance components under different management conditions

GY under MLN stress averaged 2.37 t ha$^{-1}$, with a coefficient of variation (CV) of 39.8%, indicating moderate experimental precision (Table 2). Genotypic variance ($\sigma^2_G = 0.35^{**}$) and G × E variance ($\sigma^2_{G×E} = 0.65^{**}$) for GY were highly significant ($p < 0.01$), confirming substantial genetic variability among hybrids. Variance components for MLN disease severity scores (MLN1–MLN4) were also significant ($\sigma^2_G = 0.09$–$0.52$, $p < 0.01$; $\sigma^2_{G×E} = 0.18$–$0.20$, $p < 0.01$), indicating consistent genotype differentiation across disease assessments. Mean MLN severity increased progressively from 2.0 (MLN1) to 4.28 (MLN4), confirming disease progression over time and effective inoculation pressure. Under optimum conditions, the average GY was 5.89 t ha$^{-1}$ with low CV (18.9%), demonstrating high precision. Genotypic variance for GY ($\sigma^2_G = 0.25^{**}$) and PH ($\sigma^2_G = 83.24^{**}$) were highly significant. Under drought, mean GY was 4.48 t ha$^{-1}$ with significant genetic variance. The mean ASI (0.01 days) was short, reflecting effective drought stress management and good synchronization in most genotypes.

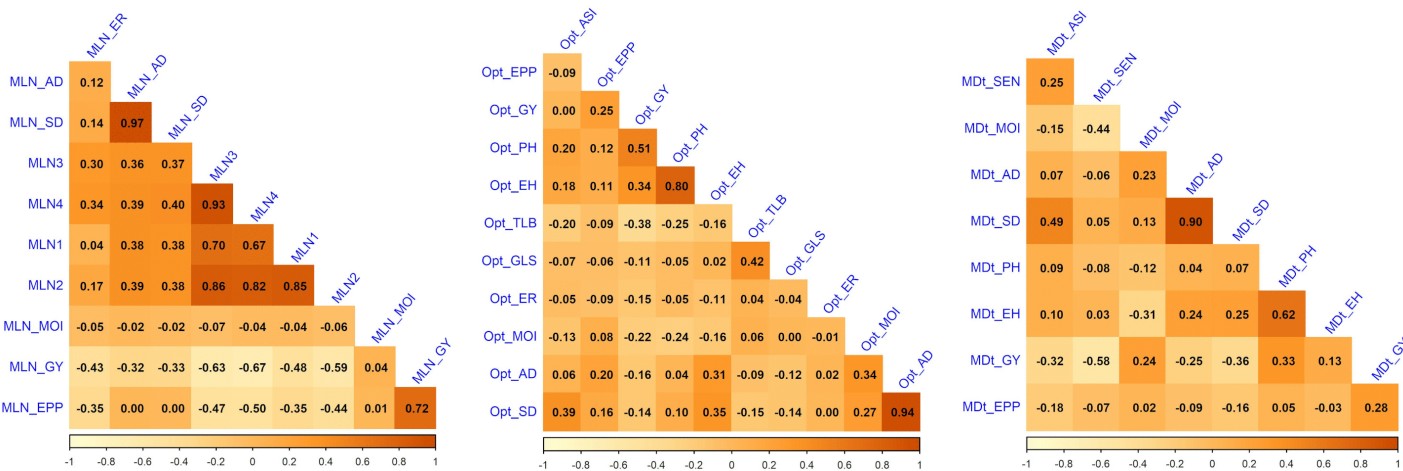

**Fig 2. Pearson's correlation between grain yield and other traits evaluated under MLN, optimum (Opt) and drought (MDt) conditions.** The correlation level is color-coded according to the color key indicated on the scale. Correlations with >0.16 were significant at 0.05 (*p*) level. AD, anthesis date; SD, silking date; GY, grain yield; PH, plant height; EH, ear height; MOI, moisture content; EPP, ears per plant; GLS, gray leaf spot; TLB, turcium leaf blight; MLN1,2,3, and 4, MLN disease severity scored at four-time intervals; ER, ear rot; SEN, senescence.

**Table 2. Estimated genetic and residual variance components for MLN disease severity, grain yield, and agronomic traits under contrasting stress and non-stress environments across locations.**

| Statistic | GY | MOI | AD | SD | EPP | MLN1 | MLN2 | MLN3 | MLN4 |
|---|---|---|---|---|---|---|---|---|---|
| **MLN disease pressure** | | | | | | | | | |
| Mean | 2.37 | 13.99 | 73.13 | 75.84 | 0.87 | 2.00 | 2.81 | 3.72 | 4.28 |
| $\sigma^2_G$ | 0.35** | 0.26** | 6.72* | 7.39* | 0.02* | 0.09* | 0.32** | 0.37** | 0.52** |
| $\sigma^2_{GxE}$ | 0.65** | 0.81** | 16.88** | 17.51** | 0.02* | 0.25** | 0.18** | 0.18** | 0.20** |
| $\sigma^2_e$ | 0.89 | 2.41 | 6.83 | 6.99 | 0.05 | 0.26 | 0.49 | 0.56 | 0.64 |
| $LSD_{5\%}$ | 2.07 | 1.27 | 5.96 | 11.61 | 0.45 | 0.68 | 0.99 | 1.05 | 1.16 |
| CV (%) | 39.84 | 11.11 | 3.57 | 3.49 | 26.72 | 25.61 | 24.95 | 20.16 | 18.72 |
| **Optimum condition** | | | | | | | | | |
| Statistic | GY | MOI | AD | SD | ASI | PH | EH | EPP | EPH |
| Mean | 5.89 | 15.42 | 63.59 | 63.65 | 0.06 | 231.10 | 120.65 | 1.01 | 0.52 |
| $\sigma^2_G$ | 0.25** | 0.17* | 3.42** | 4.12** | 0.40** | 83.24** | 66.93** | 0.001* | 0.001* |
| $\sigma^2_{GxE}$ | 0.42** | 0.68** | 0.14* | 0.22** | 0.07** | 33.01** | 21.27** | 0.001* | 0.001* |
| $\sigma^2_e$ | 1.24 | 2.71 | 2.03 | 2.31 | 0.90 | 117.40 | 78.69 | 0.01 | <0.01 |
| $LSD_{5\%}$ | 0.99 | 1.00 | 1.63 | 1.79 | 0.97 | 11.80 | 9.78 | 0.08 | 0.03 |
| CV (%) | 18.85 | 10.68 | 2.24 | 2.39 | 150.20 | 4.69 | 7.35 | 12.04 | 5.59 |
| **Managed drought** | | | | | | | | | |
| Statistic | GY | MOI | AD | SD | ASI | PH | EH | EPP | SEN |
| Mean | 4.48 | 13.55 | 62.76 | 62.78 | 0.01 | 216.88 | 118.14 | 0.95 | 2.71 |
| $\sigma^2_G$ | 0.41** | 1.58** | 2.85** | 3.84** | 0.62** | 69.81** | 91.03** | 0.001* | 0.44** |
| $\sigma^2_e$ | 0.61 | 2.06 | 1.82 | 2.41 | 0.84 | 100.48 | 71.58 | 0.01 | 0.66 |
| $LSD_{5\%}$ | 1.16 | 2.18 | 2.30 | 2.66 | 1.38 | 15.03 | 14.13 | 0.11 | 1.21 |
| CV (%) | 17.48 | 10.59 | 2.15 | 2.47 | 199.87 | 4.62 | 7.16 | 11.63 | 29.91 |

*, **, Significant at $P<0.05$; and $< 0.01$ probability levels, respectively. GY, grain yield; MOI, moisture content; AD, days to 50% anthesis; SD, Days to 50% silking; ASI, Anthesis-silking interval; PH, plant height; EH, Ear height; EPP, ears per plant; EPH, EH-PH ratio; MLN1, MLN2, MLN3 and MLN4 represents MLN disease severity at four time intervals; $\sigma^2_G$, genotypic variance; $\sigma^2_{GxE}$, LSD, least significant difference at 5%; CV, coefficient of variation.

## Mean Performance of testcross hybrids

Under MLN disease pressure, the best-performing hybrid, (CKLMARSI0037/CKLTI0139)// CKDHL120312, recorded 5.75 t ha$^{-1}$, followed by (CKLTI0139/CKLMARSI0022)// CKDHL120312 (5.24 t ha$^{-1}$) and (CKLTI0138/CKLMARSI0022)// CKDHL120312 (5.21 t ha$^{-1}$) (Fig 3, Supplementary Table S4 in S1 File). These top hybrids exhibited consistently low MLN severity scores (MLN3 ≤ 1.75; MLN4 ≤ 2.5) and high yield potential. In contrast, commercial checks such as Pioneer 3253 and DK8031 yielded ≤1.0 t ha$^{-1}$ with MLN severity scores > 5.0, confirming their susceptibility. Two MLN-resistant internal checks recorded grain yields greater than 3 t ha$^{-1}$ while maintaining MLN disease severity scores of less than 3.5. Hybrids combining CKDHL120312 as a parental line appeared most frequently among the top performers, confirming its strong

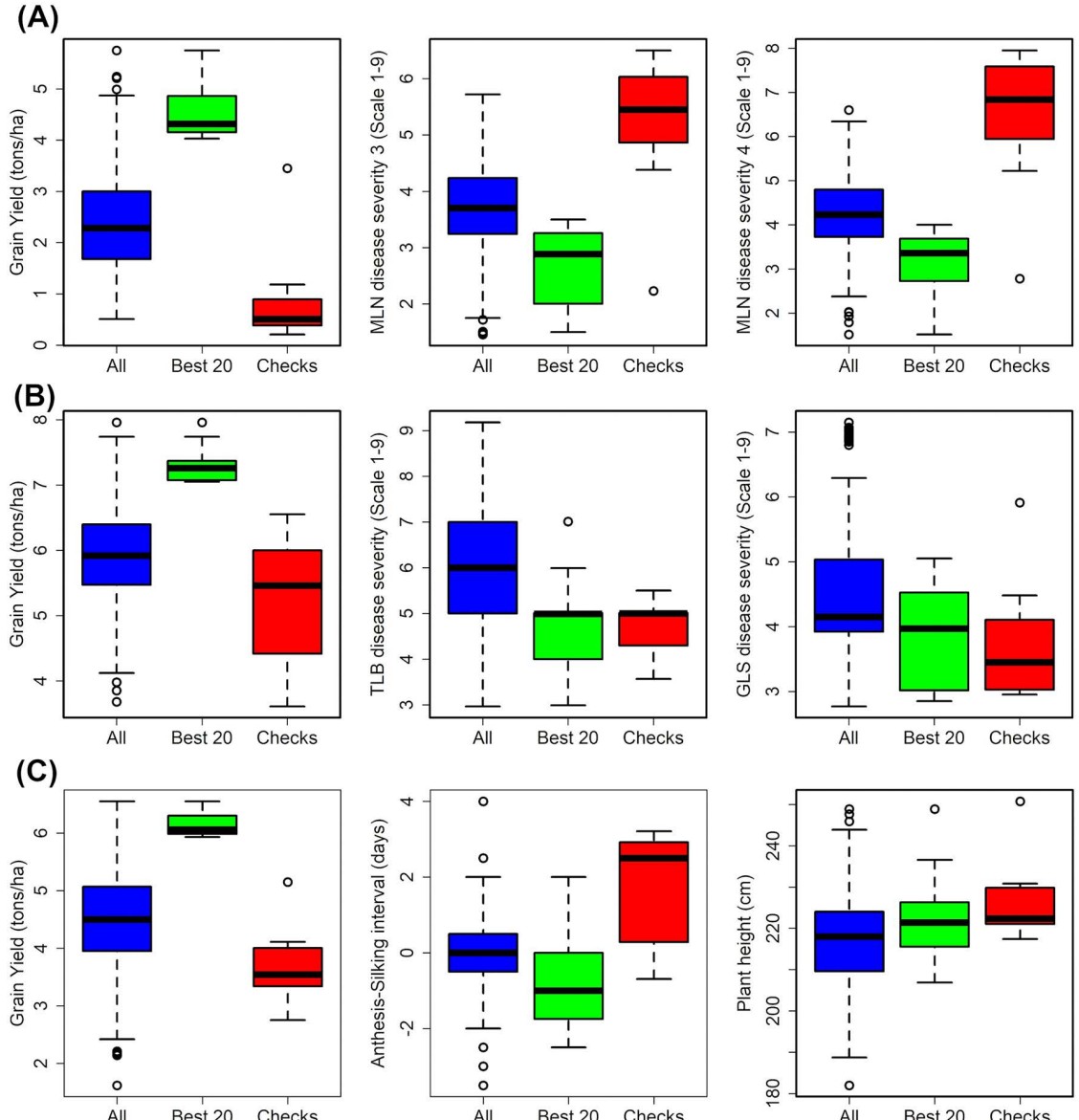

**Fig 3. Performance of all hybrids, the top 20 experimental hybrids, and commercial checks across multiple locations under MLN, optimum, and managed drought environments.**

GCA under MLN stress. The mean SD (75.8 days) suggested medium maturity, while moisture content was averaged 13.99%.

Across optimum sites, mean GY of the top 20 hybrids was 7.1 t ha⁻¹, exceeding all commercial checks by 20–30% (Fig 3, Supplementary Table S5 in S1 File). The best hybrid, (CML322/CML543)//CKDHL142445, yielded 7.96 t ha⁻¹, followed by (CKDHL120312/CML536)//CKMLN150478 (7.74 t ha⁻¹) and (CKLMARSI0037/CKLTI0139)//CKDHL140910 (7.53 t ha⁻¹). These hybrids also maintained low foliar disease scores (TLB < 5.0; GLS < 4.0), indicating durable resistance. The mean ASI was short (0.06 days), and PH averaged 231 cm, typical of high-yielding CIMMYT germplasm. Compared with checks, the elite experimental hybrids demonstrated 20–60% yield superiority, confirming successful introgression of MLN tolerance into high-performing backgrounds.

Under drought stress, mean GY of the top 20 hybrids was 5.4 t ha⁻¹, showing strong drought resilience (Fig 3, Supplementary Table S6 in S1 File). The highest-yielding hybrids included: (CKLTI0043/CKDHL120312)// CKDHL140548–5.66 t ha⁻¹, (CKSBL10194/CKDHL120312)//CKDHL140548–5.63 t ha⁻¹, (CML322/CML543)//CKDHL142445–5.60 t ha⁻¹. These hybrids displayed short ASI (≤ 1.0 day), reduced senescence (≤ 2.8 score), and moderate PH (~220 cm). The best-performing hybrids under drought shared common parents such as CKDHL140548 and CML322/CML543, emphasizing their combining ability for stress tolerance. Commercial checks yielded between 3.2–5.1 t ha⁻¹, confirming the superiority of experimental hybrids under moisture-limited conditions.

## Analysis of variance across environments

Significant mean squares were recorded for lines, testers, and L×T interactions across environments for most traits ($p < 0.01$). Under MLN stress, lines and testers contributed strongly to yield variation (Table 3). Under optimum conditions, genotypic variance for PH and EH was highly significant, indicating strong genetic differentiation for plant architecture. Under drought, lines (MS = 7.07**) and testers (MS = 3.56**) showed significant effects for GY, whereas the line×tester interaction was smaller, confirming predominance of additive gene action.

Variance partitioning showed that additive variance was consistently higher than dominance variance across management conditions (Table 4). Under MLN stress, additive variance for GY was 1.20, compared with negligible dominance variance, and heritability estimates were high (H² = 0.69; h² = 0.69). Under optimum and drought conditions, heritability for GY reached 0.79–0.88, while Baker's ratios (0.85–0.99) confirmed the reliability of GCA-based predictions. Heritability for MLN severity traits (MLN1–MLN4) ranged from 0.57 to 0.86, highlighting sufficient genetic variation for selection.

## GCA effects Across MLN, optimum, and drought environments

Substantial additive genetic variability was detected among both lines and testers across MLN, optimum, and drought environments, indicating strong genotype and GxE interactions shaping GY. The magnitude and direction of GCA effects differed widely among parents, with some showing broad adaptability while others displayed environment-specific strengths (Tables 5–7; Fig 4). Under MLN stress, several lines—including CKDHL120312, CML550, CKLTI0136, CKLTI0230, and CKLTI0134—exhibited the highest positive GCA estimates for GY, coupled with strongly negative effects for MLN severity (Table 5). CKDHL120312 was particularly outstanding, with the highest GCA for GY (+0.93) and a large negative GCA for MLN severity (−0.96 to −1.17). Among testers, CKDHL120918/CKLTI0138, CKLMARSI0037/CKLTI0139, and CKDHL120918/CKLTI0136 recorded positive GCA for yield (0.42–0.53) and reduced MLN severity (−0.16 to −0.41), confirming their value as strong male parents for MLN resistance breeding. Several parents, including CKLTI0318, CKDHL120918/CML494, and CKLMARSI0037/CML543—showed consistently negative GCA under MLN stress.

Under optimum conditions, the leading positive GCA contributors for yield included CKMLN150478 (0.37), CKDHL142445 (0.35), CKLMARSI0029 (0.31), CKLTI0139, and CKDHL140910 (Table 6). These lines also showed reduced GLS and TLB severity scores (−0.78 to −0.39), combining yield potential with foliar disease resistance. Testers such as CKDHL0221/CML464 (0.35), CML322/CML543 (0.34), and CKLMARSI0037/CML543 (0.31) were the strongest

**Table 3. Analysis of variance (ANOVA) for grain yield, disease severity, and agronomic traits across MLN, optimum, and managed drought environments.**

**MLN disease pressure**

| Source of variation | DF | GY | AD | SD | MLN1 | MLN2 | MLN3 | MLN4 |
|---|---|---|---|---|---|---|---|---|
| Env | 1 | 599.02** | 38468.37** | 40077.03** | 108.16** | 150.49** | 177.55** | 6.91** |
| Rep(Env) | 2 | 3.44* | 21.37** | 16.29 | 0.86* | 2.28** | 0.89* | 1.99** |
| Genotypes | 436 | 3.38* | 56.11** | 60.50** | 1.01* | 1.85** | 2.06** | 2.52** |
| Line | 37 | 17.81** | 312.97** | 340.36** | 2.13** | 5.32** | 7.94** | 10.83** |
| Tester | 27 | 16.90** | 159.16** | 166.77** | 10.07** | 15.52** | 14.15** | 16.66** |
| Line:Tester | 372 | 0.97* | 23.10** | 24.95** | 0.24* | 0.51 | 0.60* | 0.67* |
| Env:Genotypes | 436 | 2.19** | 43.87** | 44.54** | 0.78* | 0.85* | 0.93* | 1.04* |
| Env:Line | 37 | 11.80** | 45.50** | 33.39** | 1.13* | 1.43* | 2.14** | 2.56** |
| Env:Tester | 27 | 5.03** | 50.85** | 38.15** | 8.29** | 5.66** | 3.88** | 4.92** |
| Env:Line:Tester | 372 | 1.03** | 43.17** | 45.87** | 0.20 | 0.44 | 0.59* | 0.61* |
| Residuals | 806 | 0.86 | 5.41 | 5.31 | 0.20 | 0.46 | 0.49 | 0.56 |

**Optimum condition**

| Source of variation | DF | GY | AD | SD | PH | EH | TLB | GLS |
|---|---|---|---|---|---|---|---|---|
| Env | 2 | 398.10** | 71306.84** | 81304.65** | 359300.28** | 215942.17** | – | – |
| Rep(Env) | 3 | 0.59 | 1.90 | 9.21* | 711.68** | 227.16** | 18.09** | 2.40* |
| Genotypes | 436 | 4.04* | 17.18** | 19.56** | 748.48** | 496.00** | 3.27* | 2.30* |
| Line | 37 | 17.79** | 153.47** | 164.92** | 6145.20** | 3834.71** | 20.13** | 10.48** |
| Tester | 27 | 15.55** | 34.09** | 53.78** | 1370.92** | 1488.91** | 12.85** | 9.19** |
| Line:Tester | 372 | 1.83** | 2.40** | 2.62* | 166.64** | 91.87** | 0.90 | 0.98 |
| Env:Genotypes | 872 | 2.32** | 2.40** | 2.62* | 196.07** | 131.72** | – | – |
| Env:Line | 74 | 8.59** | 6.88** | 6.83** | 615.85** | 598.15** | – | – |
| Env:Tester | 54 | 6.67** | 5.52** | 6.51** | 740.29** | 271.46** | – | – |
| Env:Line:Tester | 744 | 1.38 | 1.72 | 1.92 | **114.81**** | 75.19 | – | – |
| Residuals | 1205 | 1.33 | 1.63 | 1.81 | 107.71 | 72.44 | 0.81 | 1.13 |

**Managed drought condition**

| Source of variation | DF | GY | AD | SD | ASI | PH | EH | MOI |
|---|---|---|---|---|---|---|---|---|
| Rep | 1 | 2.25** | 1.63* | 3.85* | 1.65* | 5316.14** | 2234.50** | 5.96* |
| Genotypes | 436 | 1.40** | 7.11** | 9.26** | 1.80* | 228.11** | 242.66** | 5.32* |
| Line | 37 | 7.07** | 52.40** | 61.09** | 8.45** | 1356.91** | 1763.95** | 29.46** |
| Tester | 27 | 3.56** | 19.79** | 36.59** | 7.04** | 337.92** | 620.35** | 18.99** |
| Line:Tester | 372 | 0.68* | 1.68** | 2.12** | 0.76* | 107.87** | 63.93* | 1.93* |
| Residuals | 403 | 0.54 | 1.42 | 1.74 | 0.63 | 100.19 | 71.98 | 1.96 |

*$P \leq 0.05$; **$P \leq 0.01$; DF, degrees of freedom; GY- grain yield; MOI- moisture content; AD- anthesis date; SD – silking date; ASI – Anthesis-silking interval; PH- plant height; EH- ear height; GLS – Gray leaf spot; TLB – Turcicum leaf blight; MLN1, MLN2, MLN3 and MLN4 corresponds to MLN disease severity at four different time intervals.

general combiners under non-stress conditions. Notably, some parents that were superior under MLN or drought, including CML550 and CKDHL120312, showed negative GCA effects under optimum conditions, highlighting environment-dependent additive gene action.

Under drought stress, the largest positive GCA effects were expressed by CKDHL140548 (1.04), CKLMARSI0029 (0.86), CKDHL140910 (0.84), CKDHL120312 (0.76), and CKSBL10194, reflecting strong additive contributions under stress conditions (Table 7). These lines also combined desirable traits such as short ASI, and moderate PH. Among

**Table 4. Line × tester variance components and Baker's ratio for grain yield and related traits across multiple locations under MLN, optimum, and managed drought environments.**

**MLN disease condition**

| Source of variation/ Trait | GY | AD | SD | MLN1 | MLN2 | MLN3 | MLN4 |
|---|---|---|---|---|---|---|---|
| Line | 0.22** | 4.75** | 5.81** | 0.01* | 0.07* | 0.09** | 0.14** |
| Tester | 0.18** | 5.00** | 5.36** | 0.04* | 0.20** | 0.21** | 0.24** |
| Line x Tester | 0.01 | 0.28** | 0.10** | 0.01* | 0.01* | <0.01 | 0.01* |
| Genotype | 0.30** | 6.39** | 6.98** | 0.06* | 0.25** | 0.29** | 0.37** |
| Additive | 1.20 | 25.56 | 27.92 | 0.23 | 1.00 | 1.14 | 1.49 |
| Dominance | <0.01 | 1.13 | 0.38 | 0.04 | 0.06 | 0.01 | 0.05 |
| Environmental | 0.55 | 10.38 | 10.73 | 0.20 | 0.21 | 0.23 | 0.26 |
| Broad Heritability | 0.69 | 0.72 | 0.73 | 0.57 | 0.83 | 0.83 | 0.86 |
| Narrow Heritability | 0.69 | 0.69 | 0.72 | 0.49 | 0.79 | 0.83 | 0.83 |
| Baker's ratio | 0.99 | 0.99 | 0.99 | 0.92 | 0.97 | 0.99 | 0.98 |

**Optimum condition**

| Source of variation/ Trait | GY | AD | SD | PH | EH | TLB | GLS |
|---|---|---|---|---|---|---|---|
| Line | 0.13** | 3.10** | 3.01** | 75.06** | 44.10** | 0.78** | 0.42** |
| Tester | 0.09* | 0.47* | 0.82** | 7.09* | 14.34** | 0.43** | 0.29** |
| Line x Tester | 0.08* | 0.17** | 0.17** | 8.77** | 2.80** | 0.04 | 0.00 |
| Genotype | 0.29** | 3.73** | 4.26** | 94.91** | 61.25** | 1.26** | 0.60** |
| Additive | 1.16 | 14.93 | 17.03 | 379.65 | 244.98 | 5.03 | 2.41 |
| Dominance | 0.31 | 0.68 | 0.69 | 35.07 | 11.18 | 0.17 | 0.00 |
| Environmental | 0.39 | 0.40 | 0.43 | 32.82 | 21.91 | 0.41 | 0.54 |
| Broad Heritability | 0.79 | 0.98 | 0.98 | 0.93 | 0.92 | 0.93 | 0.82 |
| Narrow Heritability | 0.62 | 0.93 | 0.94 | 0.85 | 0.88 | 0.90 | 0.82 |
| Baker's ratio | 0.85 | 0.98 | 0.98 | 0.95 | 0.98 | 0.98 | 1.00 |

**Managed Drought**

| Source of variation/ Trait | GY | AD | SD | ASI | PH | EH | MOI |
|---|---|---|---|---|---|---|---|
| Line | 0.26** | 2.48** | 2.48** | 0.29** | 68.92** | 72.94** | 0.56* |
| Tester | 0.10* | 0.60* | 1.12** | 0.20** | 8.32** | 19.58** | 0.91* |
| Line x Tester | 0.07* | 0.12* | 0.18* | 0.06* | 3.53** | <0.01 | <0.01 |
| Genotype | 0.43** | 2.90** | 3.85** | 0.58** | 65.37** | 87.29** | 1.67* |
| Additive | 1.71 | 11.61 | 15.39 | 2.33 | 261.48 | 349.14 | 6.69 |
| Dominance | 0.27 | 0.46 | 0.74 | 0.25 | 14.11 | <0.01 | <0.01 |
| Environmental | 0.27 | 0.73 | 0.88 | 0.31 | 50.25 | 33.73 | 0.97 |
| Broad Heritability | 0.88 | 0.94 | 0.95 | 0.89 | 0.85 | 0.91 | 0.87 |
| Narrow Heritability | 0.76 | 0.91 | 0.90 | 0.81 | 0.80 | 0.91 | 0.87 |
| Baker's ratio | 0.91 | 0.98 | 0.98 | 0.94 | 0.98 | 0.99 | 0.99 |

*, **, Significant at $P < 0.05$; and $< 0.01$ probability levels, respectively. GY- grain yield; MOI- moisture content; AD- anthesis date; SD – silking date; ASI – Anthesis-silking interval; PH- plant height; EH- ear height; GLS – Gray leaf spot; TLB – Turcicum leaf blight; MLN1, MLN2, MLN3 and MLN4 corresponds to MLN disease severity at four different time intervals.

testers, CKDHL120918/CKLMARSI0029 (0.46), CML322/CML543 (0.38), and CKSBL10194/CKDHL120312 (0.37) were the most promising drought combiners, showing high yield potential and efficient moisture use, supported by reduced senescence scores.

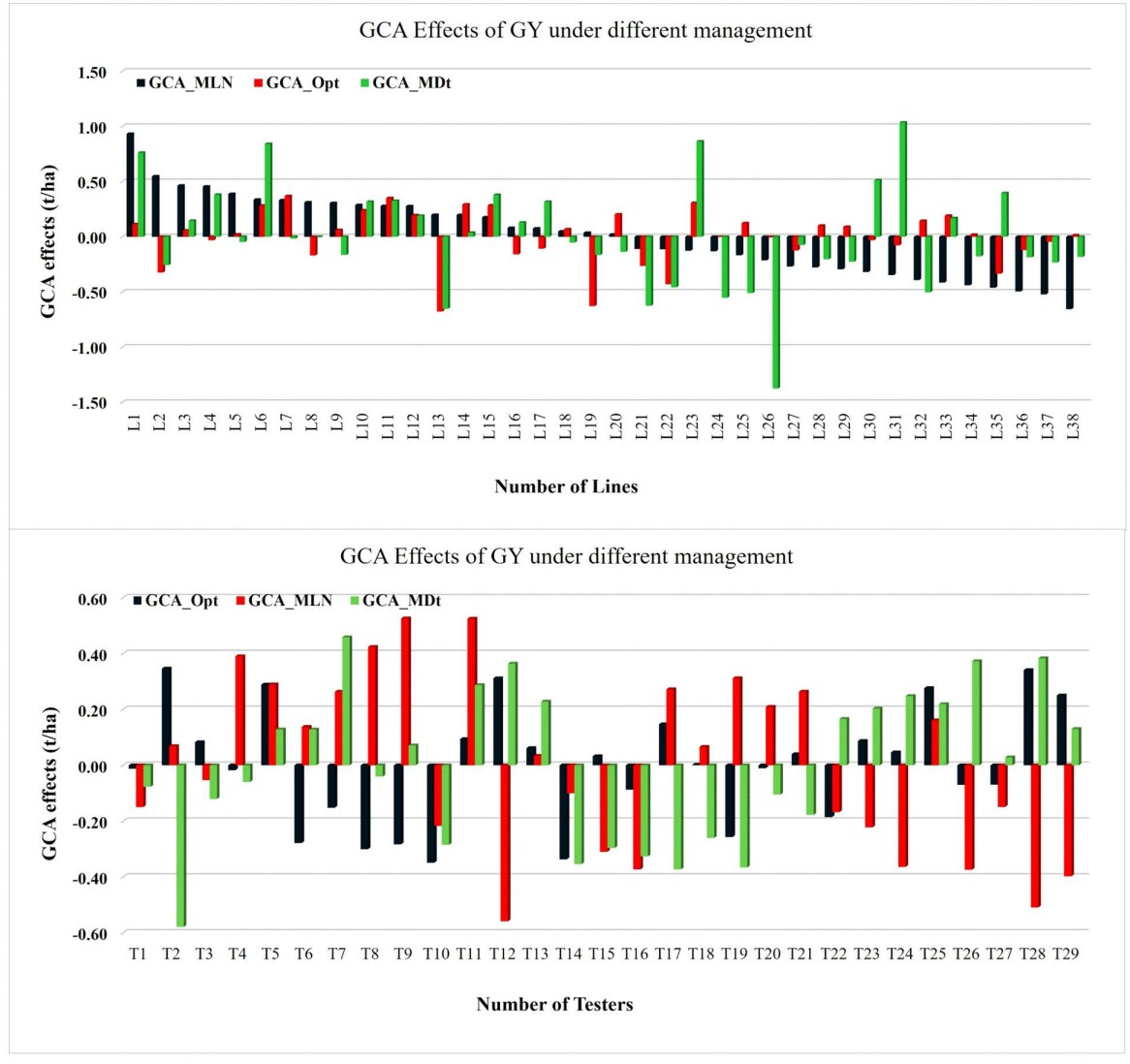

**Fig 4. General combining ability (GCA) estimates for grain yield evaluated under MLN, optimum, and drought conditions across multiple locations.**

Across management, a few parents emerged as broadly favorable general combiners. Among lines, CKDHL120312 and CKDHL140910 showed consistently positive GCA effects across MLN, optimum, and drought environments, making them strong candidates for heterotic pool improvement and multi-environment breeding pipelines (Fig 4). The lines CKLTI0230 and CKDHL142445 also showed favorable GCA in at least two management conditions. Among testers, CKDHL120918/CKLMARSI0029, CKLMARSI0037/CKLTI0139, and CKDHL120312/CML536 demonstrated stable positive contributions across management conditions. In contrast, CKDHL120918, CML495, CKLTI0137/CKLTI0330, and several CML543-based testers consistently exhibited negative GCA effects across environments.

## Specific combining ability effects of the line × tester crosses

Significant variability was observed for GY among testcross hybrids for SCA effects under both optimum and drought conditions, indicating the presence of substantial non-additive gene action influencing GY performance across environments

**Table 5. General combining ability (GCA) effects of the top 20 lines and five testers for grain yield, MLN severity, and associated agronomic traits.**

| Line | GY (t/ha) | AD (days) | SD (days) | MLN1 (1–9) | MLN2 (1–9) | MLN3 (1–9) | MLN4 (1–9) | GYOpt (t/ha) | GY Dt (t/ha) |
|---|---|---|---|---|---|---|---|---|---|
| CKDHL120312 | 0.93** | −4.51** | −4.98** | −0.18* | −0.63** | −0.96** | −1.17** | 0.11 | 0.76** |
| CML550 | 0.55** | −3.02** | −3.02** | 0.04 | −0.04 | −0.15 | −0.17 | −0.32 | −0.26 |
| CKLTI0136 | 0.46 | 0.63 | 1.23 | −0.01 | 0.02 | 0.02 | −0.05 | 0.06 | 0.15 |
| CKLTI0230 | 0.46 | −1.95 | −2.40** | −0.04 | −0.12 | −0.08 | −0.15 | −0.03 | 0.38* |
| CKLTI0134 | 0.39 | −0.48 | −0.54 | −0.01 | −0.03 | −0.01 | 0.01 | 0.02 | −0.05 |
| CKDHL140910 | 0.34 | 1.05 | 1.09 | 0.04 | 0.04 | −0.02 | −0.08 | 0.28 | 0.84** |
| CKMLN150478 | 0.33 | −0.42 | −0.53 | −0.12 | −0.14 | −0.06 | −0.08 | 0.37 | −0.02 |
| CKDHL141105 | 0.31 | 3.25** | 3.53** | 0.00 | −0.03 | 0.00 | 0.05 | −0.17 | 0.00 |
| CML494 | 0.31 | 1.03 | 0.92 | −0.04 | −0.18 | −0.14 | −0.19 | 0.06 | −0.16 |
| CKDHL140700 | 0.29 | 1.12 | 1.81 | 0.13 | 0.21 | 0.10 | 0.02 | 0.24 | 0.32 |
| CKDHL142445 | 0.28 | 0.81 | 0.97 | 0.03 | 0.12 | 0.20 | 0.09 | 0.35 | 0.33 |
| CKLMLN150478 | 0.28 | −0.43 | 0.02 | −0.06 | −0.10 | −0.06 | −0.04 | 0.20 | 0.19 |
| CKDHL120918 | 0.20 | −1.34 | −2.40** | −0.04 | −0.33* | −0.50** | −0.55 | −0.68** | −0.65** |
| CKLTI0139 | 0.20 | −0.06 | −0.03 | 0.02 | 0.04 | 0.01 | −0.03 | 0.29 | 0.04 |
| CKLMLN150474 | 0.18 | −0.52 | −0.71 | −0.14 | −0.29* | −0.21 | −0.24 | 0.28 | 0.38* |
| CKLTI0133 | 0.08 | −1.03 | −1.02 | 0.03 | 0.04 | 0.07 | 0.03 | −0.16 | 0.13 |
| CKDHL142806 | 0.08 | 1.60** | 1.73** | −0.03 | −0.09 | −0.06 | −0.05 | −0.11 | 0.32 |
| CKLMLN150459 | 0.05 | −2.04 | −2.84** | −0.12 | −0.28* | −0.23 | −0.39* | 0.07 | −0.05 |
| CKDHL143607 | 0.04 | −1.05 | −1.21 | −0.04 | −0.11 | −0.16 | −0.24 | −0.63** | −0.17 |
| CKLTI0043 | 0.02 | 2.12** | 2.16** | 0.01 | 0.24 | 0.30 | 0.29 | 0.20 | −0.14 |
| **Tester** | | | | | | | | | |
| CKDHL120918/CKLTI0138 | 0.53* | −1.43 | −1.33 | −0.07 | −0.26 | −0.29 | −0.41 | −0.28 | 0.07 |
| CKLMARSI0037/CKLTI0139 | 0.52* | −0.42 | −0.73 | −0.05 | −0.22 | −0.16 | −0.24 | 0.09 | 0.29* |
| CKDHL120918/CKLTI0136 | 0.42 | −1.37 | −1.41 | −0.07 | −0.34 | −0.24 | −0.31 | −0.30 | −0.04 |
| CKDHL120312/CML312 | 0.39 | −2.08 | −2.22** | −0.06 | −0.22 | −0.26 | −0.30 | −0.02 | −0.06 |
| CKLTI0139/CKDHL120918 | 0.31 | −2.08 | 1.36 | −0.08 | −0.43 | −0.66** | −0.64** | −0.26 | −0.37* |

*, **, Significant at P < 0.05; and < 0.01 probability levels, respectively, GY- grain yield; AD- anthesis date; SD – silking date; ASI – Anthesis-silking interval; MLN1, MLN2, MLN3 and MLN4 corresponds to MLN disease severity at four different time intervals; GY Opt – GY under optimum; GY MDt – GY under managed drought.

(Supplementary Table S7 in S1 File). The magnitude and direction of SCA effects varied widely among line × tester combinations, highlighting specific parental combinations that expressed superior hybrid performance due to favorable allelic interactions.

Under optimum conditions, the highest positive SCA effects were recorded for hybrids CML495 × CKLMARSI0037/CML543, CKLMARSI0029 × CKLTI0133/CKDHL120312, and CKDHL142425 × CKLMARSI0037/CML543, with SCA values of 0.41, 0.36, and 0.33, respectively. Several combinations involving CKDHL142425 with testers containing CML543 or CML322 consistently ranked among the top-performing crosses, suggesting strong heterotic complementation and favorable dominance interactions under favorable growing conditions.

Under drought stress, a different set of hybrids exhibited superior SCA performance. The highest drought-specific SCA values were observed for CKDHL141105 × CKDHL120918/CML494 (0.39), followed by CML550 × CKLTI0133/CKDHL120312 (0.29) and CKLMLN150356 × CKLTI0133/CKDHL120312 (0.27). These results indicate that drought tolerance is influenced by stress-specific gene interactions, with certain L × T combinations expressing enhanced dominance

**Table 6. General combining ability (GCA) effects of the top 20 lines and five testers for grain yield and agronomic traits under optimum conditions.**

| Line | GY | AD | SD | PH | EH | TLB | GLS |
|---|---|---|---|---|---|---|---|
| CKMLN150478 | 0.37 | −1.18** | −0.86 | 6.45* | 2.59 | −0.11 | 0.47* |
| CKDHL142445 | 0.35 | 2.72** | 2.50** | 15.35** | 11.39** | −0.44 | −0.39 |
| CKLMARSI0029 | 0.31 | −1.19** | −0.80 | 8.45** | −4.65 | −1.03** | −0.78** |
| CKLTI0139 | 0.29 | 0.31 | 0.39 | 10.97** | 7.96** | −0.14 | 0.07 |
| CKLMLN150474 | 0.28 | −1.04* | −0.94* | 1.27 | −1.87 | −0.95** | −0.29 |
| CKDHL140910 | 0.28 | 1.08** | −0.37 | −6.66** | −2.36 | −1.11** | −0.80** |
| CKDHL140700 | 0.24 | 2.55** | 2.04** | 13.16** | 4.15 | 0.21 | 0.03 |
| CKLTI0043 | 0.20 | 1.76** | 1.82** | −2.92 | −2.31 | −1.20** | −0.52* |
| CKLMLN150478 | 0.20 | −1.14** | −0.44 | 4.02 | −0.73 | 0.06 | 0.34 |
| CKDHL120358 | 0.19 | −0.10 | 0.22 | −3.74 | −5.75** | 0.02 | 0.22 |
| CKDHL140539 | 0.14 | 2.00** | 1.52** | 2.25 | 5.88** | −0.21 | 1.73** |
| CKDHL120668 | 0.12 | −1.27** | −0.95* | 2.26 | −3.61 | −1.12** | −0.89** |
| CKDHL120312 | 0.11 | −2.74** | −3.30** | −3.84 | −6.94** | 0.90** | 0.63* |
| CKLMARSI0022 | 0.10 | −1.04* | −0.74 | −0.89 | −3.95 | −0.95** | −0.93** |
| CKLMLN150340 | 0.09 | −0.95* | −0.43 | 5.32 | 5.61** | −1.23** | −0.88** |
| CKLMLN150459 | 0.07 | −1.33** | −1.52** | −4.48 | −3.42 | 0.35 | 0.07 |
| CML494 | 0.06 | 0.58 | 0.73 | 8.60** | 8.37** | 0.25 | 0.08 |
| CKLTI0136 | 0.06 | 0.94* | 1.38** | 9.99** | 7.69** | −0.15 | −0.16 |
| CKLTI0134 | 0.02 | −0.27 | −0.41 | 5.40 | 4.59 | −0.28 | 0.28 |
| CKLMLN150356 | 0.02 | −4.81** | −4.59** | −13.70** | −11.19** | 0.13 | 0.32 |
| Tester | | | | | | | |
| CKDHL0221/CML464 | 0.35 | 1.66** | 2.40** | 4.15** | 10.47** | −1.83** | −1.04** |
| CML322/CML543 | 0.34 | −0.06 | −0.03 | 0.70 | 1.64 | −1.47** | 0.35 |
| CKLMARSI0037/CML543 | 0.31 | −0.44 | 0.25 | 0.59 | 0.72 | −0.94** | 0.09 |
| CKDHL120312/CML536 | 0.29 | −0.16 | −0.34 | 2.07 | −0.02 | −0.89** | −0.54* |
| CKSBL10060/CKDHL120312 | 0.28 | −1.52** | −1.77** | −0.44 | −1.01 | −0.13 | −0.05 |

*, **, Significant at P<0.05; and < 0.01 probability levels, respectively, GY- grain yield; AD- anthesis date; SD – silking date; PH- plant height; EH- ear height; GLS – Gray leaf spot; TLB – Turicum leaf blight.

or over-dominance effects specifically under low-moisture conditions. Hybrids involving tester CKLTI0133/CKDHL120312 appeared multiple times among the top drought performers, underscoring this tester's strong combining ability under water-deficit stress.

A few hybrids demonstrated consistently favorable SCA under both optimum and drought conditions. Notably, CKDHL142425 × CML322/CML543 and CML550 × CKLTI0133/CKDHL120312 showed high and positive SCA effects across environments, suggesting the presence of robust and stable non-additive interactions contributing to broad adaptation. In contrast, several hybrids showed large negative SCA values under both optimal and drought conditions, indicating poor specific complementation between the respective parental lines. These combinations are less likely to produce competitive hybrids. Overall, the contrasting sets of top-performing hybrids across environments reinforce the importance of evaluating breeding materials under different management conditions. The results highlight clear opportunities to exploit non-additive genetic effects to enhance hybrid productivity and stability, particularly through strategic use of testers such as CML543-, CML322-, and CKLTI0133-based combinations, which repeatedly contributed to superior performance.

**Table 7. General combining ability (GCA) effects of the top 20 lines and five testers for grain yield and agronomic traits under managed drought conditions.**

| Lines | GY | AD | SD | ASI | PH | EH | MOI |
|---|---|---|---|---|---|---|---|
| CKDHL140548 | 1.04** | 0.95** | −0.24 | −1.17** | −3.18 | 10.27** | 0.02 |
| CKLMARSI0029 | 0.86** | −0.81* | −0.73* | −0.02 | 9.62** | −6.09** | −0.12 |
| CKDHL140910 | 0.84** | 0.33 | −1.33** | −1.50** | −2.66 | −2.17 | 0.67* |
| CKDHL120312 | 0.76** | −2.54** | −3.63** | −0.94** | 4.24** | −7.01** | 0.77* |
| CKSBL10194 | 0.51* | −0.55* | −0.05 | 0.60* | 0.41 | −0.92 | 0.43 |
| CKDHL140475 | 0.40* | −1.42** | −1.78** | −0.25 | 3.09 | −5.99* | 1.33* |
| CKLTI0230 | 0.38* | −0.91** | −0.46 | 0.35 | 3.45 | 6.52** | 0.01 |
| CKLMLN150474 | 0.38 | −1.46** | −1.27** | 0.03 | 1.25 | −4.78* | −0.06 |
| CKDHL142445 | 0.33 | 2.58** | 2.24** | −0.08 | 20.23** | 11.76** | 0.86* |
| CKDHL140700 | 0.32 | 2.86** | 2.11** | −0.44 | 18.55** | 4.56 | 0.64* |
| CKDHL142806 | 0.32 | 1.60** | 0.94** | −0.50* | −8.85** | −9.50** | −1.03* |
| CKLMLN150478 | 0.19 | −1.59** | −1.16 | 0.25 | 7.82** | 12.53** | −1.27* |
| CKDHL120358 | 0.17 | −0.55 | 0.21 | 0.81* | 1.59 | −10.96** | −0.71 |
| CKLTI0136 | 0.15 | 0.69 | 1.29** | 0.50* | 3.58 | 7.93** | −0.19 |
| CKLTI0133 | 0.13 | −0.06 | 0.39 | 0.33 | 6.52** | 7.69** | −0.41 |
| CKLTI0139 | 0.04 | −0.66 | −0.67 | −0.07 | 1.03 | 3.70 | −0.03 |
| CKDHL141105 | 0.00 | 2.52** | 2.81** | 0.40* | 4.29** | 9.65** | 0.15 |
| CKMLN150478 | −0.02 | −0.01 | 0.06 | −0.01 | 3.99 | 4.32 | 0.04 |
| CKLTI0134 | −0.05 | −0.24 | −0.19 | −0.02 | −0.97 | 5.75* | −0.01 |
| CKLMLN150459 | −0.05 | −1.43** | −1.54** | −0.21 | −7.63** | −6.54** | 0.65* |
| Testers | | | | | | | |
| CKDHL120918/CKLMARSI0029 | 0.46* | 0.07 | −0.03 | −0.15 | −1.16 | −6.23** | 1.83* |
| CML322/CML543 | 0.38* | −0.29 | −0.34 | −0.11 | 1.89 | 2.79 | 0.50 |
| CKSBL10194/CKDHL120312 | 0.37* | −0.49 | −0.77** | −0.13 | 0.62 | 4.71* | −1.17* |
| CKLMARSI0037/CML543 | 0.36* | 0.05 | 0.83** | 0.63** | 3.32* | 2.90 | 0.54 |
| CKLMARSI0037/CKLTI0139 | 0.29* | 0.01 | 0.22 | 0.12 | −1.08 | 0.78 | 0.86* |

*, **, Significant at P<0.05; and < 0.01 probability levels, respectively, GY- grain yield; AD- anthesis date; SD – silking date; ASI – Anthesis-silking interval; PH- plant height; EH- ear height; MOI – Moisture content

## Visualization of the line-by-testers' relationship among inbred lines and single cross testers

The interrelationships among the 38 inbred lines and 29 single-cross testers were examined using a GGE biplot to enable intuitive graphical interpretation (Fig 5). The GGE biplots were generated separately for each management condition using the mean GY of the L×T matrix as input. The GGE biplots explained 86.24% of the total variation in the L×T matrix under MLN conditions, 77.89% under optimum conditions, and 76.80% under drought stress, providing sufficient power for meaningful comparisons and reliable interpretation of line–tester relationships. The discriminating ability and representativeness of the testers varied across environments, as clearly visualized in the biplots (Fig 5). Overall, the testers exhibited distinct differences in both their ability to discriminate among inbred lines and their representativeness across the three management conditions.

## Discussion

A resilient and forward-looking hybrid breeding program relies on a broad and constantly evolving germplasm base [33]. Sustained genetic improvement requires not only the regular infusion of novel alleles but also the systematic

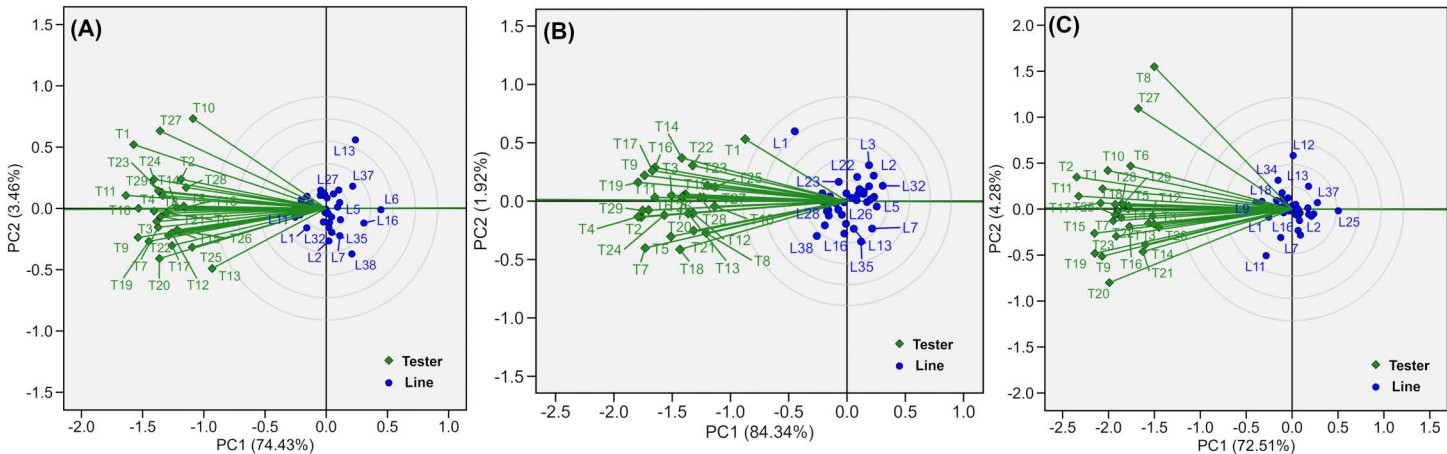

**Fig 5. GGE biplot displaying relationship among testers for heterotic grouping across locations under the management of (A) MLN, (B) Optimum, and (C) drought conditions scaled (divided) by: 0 = no-scaling; centered by:2 = tester-centered, G + GE; S.V.P. (singular value partitioning): 2 tester-metric preserving or tester-focused or column metric preserving.**

characterization and structuring of both new and established germplasm according to their combining ability and heterotic relationships [6]. Because the extent and organization of genetic diversity ultimately shape heterotic patterns, understanding these two components is fundamental for designing efficient breeding pipelines. Accordingly, knowledge of diversity and heterotic group structure forms the backbone of successful hybrid development and has consistently been highlighted as a key determinant of long-term breeding progress. The present study leveraged a large and diverse panel of 38 inbred lines and 29 single-cross testers from CIMMYT's elite maize breeding pipelines to dissect combining ability patterns and hybrid performance across MLN, optimum, and drought stress conditions. The germplasm encompassed MLN-tolerant, drought-resilient, and insect-resistant genetic backgrounds (Table 1). The breadth of allelic diversity embedded in these materials provided a robust platform to capture additive and non-additive genetic effects, enabling a comprehensive assessment of hybrid potential under contrasting regimes.

### Trait relationships and genetic architecture across environments

The continuous distributions observed for GY and other traits across testing locations reaffirm the quantitative nature of these traits and the substantial polygenic variation available for selection (Fig 1). Strong positive correlations between GY and EPP, as well as negative associations between MLN severity and yield, validate key physiological pathways contributing to productivity under both disease and abiotic stress (Fig 2). The weak but consistent negative correlation between ASI and yield under drought further confirms ASI as a reliable stress-adaptation indicator, in line with earlier findings from tropical maize improvement programs [9,34].

Highly significant genotypic and GxE interaction variances for GY and MLN severity demonstrate substantial exploitable variability. The high mean MLN severity observed across progressive scoring stages confirmed the reliability of inoculation pressure and ensured robust differentiation among hybrids. The significant variance components for yield and other agronomic traits under optimum conditions highlight the potential for further genetic gains when selection is conducted under high-input management. Under drought stress, the moderate reduction in mean GY (4.48 t ha$^{-1}$) combined with significant genetic variance highlights the potential for selecting drought-tolerant lines. The short ASI observed under stress reflects well-managed drought imposition and good reproductive resilience among the evaluated genotypes, supporting the reliability of stress phenotyping.

## Hybrid performance and trait responses under stress and non-stress conditions

The superior performance of experimental hybrids relative to commercial checks across MLN, optimum, and drought environments emphasizes the effectiveness of recent germplasm improvement efforts (Tables 3–5 to 5). Under MLN stress, hybrids involving CKDHL120312 and MLN-tolerant single-cross testers such as CKLMARSI0037/CKLTI0139 produced up to fivefold higher yields than commercial checks, demonstrating successful integration of MLN resistance alleles (Fig 3, Supplementary Table S4 in S1 File). The top-performing MLN hybrids consistently exhibited low disease scores, confirming the value of combining physiological tolerance.

Across optimum locations, yield advantages of 20–60% over commercial checks indicate that introgressing MLN tolerance into high-yielding, foliar disease–resistant backgrounds did not compromise agronomic performance (Fig 3, Supplementary Table S5 in S1 File). Under drought stress, elite hybrids combining CML322/CML543 with CKDHL140548 or CKDHL120312-derived lines showed strong yield resilience supported by short ASI, delayed senescence, and robust plant vigor—traits that underpin drought adaptation in tropical maize. Collectively, the consistent superiority of experimental hybrids across environments highlights the success of integrating MLN resistance, drought tolerance, and high-yield potential in a unified breeding pipeline.

## Additive and non-additive gene action shaping hybrid performance

Variance partitioning revealed that additive genetic effects were the dominant source of variation across management. High heritability estimates (0.69–0.88) and large Baker's ratios (0.85–0.99) underscore the reliability of selecting parents based on GCA and highlight the efficiency of forward breeding strategies for these traits. The strong additive variance for MLN severity further suggests that recurrent selection and GCA-driven parental improvement will accelerate gains in MLN resistance. Similar predominance of additive variance for MLN severity traits was also observed in earlier studies [25,35].

The significant SCA effects observed across optimum and drought conditions indicate that non-additive gene action also plays a role in the expression of GY and key agronomic traits (Supplementary Table S7 in S1 File). Only a small subset of hybrids expressed consistently high and positive SCA effects within management conditions, suggesting that specific parental combinations exploit heterosis more efficiently than others and that heterotic complementation under stress is highly genotype dependent.

Under optimum conditions, hybrids with strong positive SCA for yield and yield-contributing traits demonstrated robust combining patterns arising from favorable dominance and epistatic interactions. In contrast, the limited number of hybrids showing significant positive SCA under drought stress highlights the stringent nature of water-limited environments, where only combinations with superior stress-adaptive allelic interactions achieve enhanced performance. This emphasizes the importance of testing hybrid combinations in both managed stress and non-stress conditions to accurately identify crosses with across environment or environment-specific heterotic advantages.

The contrasting SCA patterns between optimum and drought environments further illustrate GxE-dependent expression of non-additive effects. Hybrids performing well exclusively under drought likely carry stress-responsive alleles that interact favorably when combined, whereas those performing under both conditions likely possess more stable dominance complementation. These findings reinforce the value of L×T designs for identifying elite parental combinations and provide a genetic basis for advancing top-performing hybrids into multi-location testing. Overall, the results confirm that non-additive gene action is also critical in hybrid performance, particularly under stress, and that targeted parental selection based on SCA effects can accelerate the development of high-yielding, drought-resilient maize hybrids suitable for SSA.

## Insights From Line×Tester combining ability across management

Across MLN-stressed environments, only a small subset of lines exhibited favorable GCA, combining positive effects on grain yield with negative effects on MLN severity. This scarcity highlights the challenge of simultaneously improving yield and disease resistance and emphasizes the importance of these lines as donor parents for MLN resistance breeding. Under optimum conditions, few lines showed high GCA for both yield and foliar disease resistance, indicating strong

additive genetic effects and their suitability for heterotic pool improvement. Under drought stress, CKDHL140548 and CKLMARSI0029 consistently contributed favorable additive effects, identifying them as key sources of drought resilience.

Among testers, a limited number of elite combinations expressed stable and favorable GCA across environments, reflecting a robust additive genetic architecture and their value in identifying broadly adapted hybrids. Overall, tester GCA patterns indicate that additive effects play a major role in determining hybrid performance. Under MLN pressure, only a few testers combined positive GCA for yield with reduced disease severity, whereas others showed unfavorable effects. Under optimum conditions, few testers were strong general combiners, while some stress-adapted parents performed poorly. Collectively, these results demonstrate that environment-specific GCA patterns are critical for parental selection and highlight the need to target additive genetic effects to develop maize hybrids with stable performance across contrasting stress and non-stress environments.

The choice of a suitable tester in hybrid breeding programs depends on its ability to effectively discriminate among new maize inbred lines for target traits. In this study, the significant L×T interactions for GY and related agronomic traits across management conditions highlight the differential effectiveness of testers in identifying MLN-resistant and high-yielding inbreds. Across environments and management conditions, testers such as CKDHL120918/CKLMARSI0029, CKLMARSI0037/CKLTI0139, and CKDHL120312/CML536 consistently showed positive and stable GCA effects, indicating their broad utility in hybrid development. In contrast, few testers exhibited consistently negative GCA, reducing their suitability as donor parents. These findings align with earlier reports demonstrating that tester performance varies with genetic background and test environment [36–40]. The integration of GCA effects, magnitude of genetic variance, favorable allele frequencies, and mean testcross performance remains critical for selecting robust testers [41,42]. Overall, the GCA patterns reveal both broadly adaptive and environment-specific additive effects, providing a strong framework for strategic tester deployment. Prioritizing testers with stable GCA for MLN resistance, high yield under optimum conditions, and strong drought tolerance will enhance selection efficiency and accelerate the development of resilient maize hybrids for SSA.

The GGE biplots effectively summarized relationships among lines and testers under MLN, optimum, and drought conditions, explaining 76–86% of the variation (Fig 5). The differential discriminating ability and representativeness of testers across environments confirm that no single tester can reliably profile all lines, particularly under stress. However, few testers not only showed positive and stable GCA effects and consistently provided both strong discrimination and good representativeness, validating their use in multi-environment selection pipelines.

## Implications for breeding and deployment

The combined evidence from GCA, SCA, hybrid performance, heritability, and GGE biplots clearly demonstrates that: (i) additive gene action predominantly drives MLN resistance, and other agronomic traits supporting continued use of genomic selection, forward breeding, and recurrent selection, (ii) non-additive gene action is also critical for hybrid performance under stress, especially drought, and should be exploited through targeted L×T mating designs, and (iii) clustered sets of strong combiners (e.g., CKDHL120312, CKDHL140548, CKLMARSI0037/CKLTI0139, CML322/CML543) offer a strategic foundation for building heterotic patterns tailored to MLN-endemic and drought-prone environments. Hybrid development integrating MLN resistance, disease tolerance, and drought resilience is achievable without compromising yield potential, as demonstrated by the consistently superior experimental hybrids. Overall, this study provides a comprehensive framework for deploying elite MLN-resistant, high-yielding, and drought-tolerant hybrids in eastern Africa and contributes to the global understanding of combining ability and hybrid development under multiple stress environments.

## Supporting information

**S1 File. S1 Table.** BLUEs and BLUPs for 437 testcross hybrids and ten commercial checks evaluated under MLN disease pressure. **S2 Table.** BLUEs and BLUPs for 437 testcross hybrids and ten commercial checks evaluated under optimum conditions. **S3 Table.** BLUEs and BLUPs for 437 testcross hybrids and ten commercial checks evaluated undermanaged drought conditions. **S4 Table.** Mean performance of grain yield, MLN disease severity at different time intervals and other

agronomic traits in best 20 testcross hybrids plus seven checks tested in multiple locations under MLN disease conditions. **S5 Table.** Mean performance of grain yield, and other agronomic traits in best 20 testcross hybrids plus seven checks tested in multiple locations under optimum conditions. **S6 Table.** Mean performance of grain yield, and other agronomic traits in best 20 testcross hybrids plus seven checks tested under managed drought conditions and their performance for GLS, TLB and GY under optimum and for MLN and GY under MLN disease pressure conditions. **S7 Table.** Specific combining ability effects of 437 testcross hybrids for grain yield under MLN disease pressure, optimum and managed drought conditions.
(ZIP)

## Acknowledgments

The authors thank CIMMYT scientists and the technical team in Kenya for their support. We extend our gratitude to the management of the Kenya Agricultural and Livestock Research Organization (KALRO) stations for allowing access to experimental facilities across various locations in Kenya, where field experiments were conducted and KALRO staff for aiding data collection.

## Author contributions

**Conceptualization:** Manje Gowda, Yoseph Beyene.

**Data curation:** Suresh Lingadahalli Mahabaleswara, Manigben Kulai Amadu.

**Formal analysis:** Veronica Ogugo, Manigben Kulai Amadu.

**Funding acquisition:** Manje Gowda, Yoseph Beyene.

**Methodology:** Suresh Lingadahalli Mahabaleswara.

**Project administration:** Manje Gowda, Yoseph Beyene, Vijay Chaikam.

**Resources:** Manje Gowda, Vijay Chaikam.

**Software:** Suresh Lingadahalli Mahabaleswara, Veronica Ogugo, Vijay Chaikam.

**Supervision:** Manje Gowda, Yoseph Beyene.

**Validation:** Suresh Lingadahalli Mahabaleswara, Manigben Kulai Amadu.

**Visualization:** Manigben Kulai Amadu.

**Writing – original draft:** Manje Gowda.

**Writing – review & editing:** Manje Gowda, Yoseph Beyene, Suresh Lingadahalli Mahabaleswara, Veronica Ogugo, Manigben Kulai Amadu, Vijay Chaikam.

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
