## [Decision Letter · Decision Letter 0]

26 Dec 2025

Dear Dr. Gowda,

Thank you for submitting your manuscript to PLOS ONE. After careful consideration, we feel that it has merit but does not fully meet PLOS ONE’s publication criteria as it currently stands. Therefore, we invite you to submit a revised version of the manuscript that addresses the points raised during the review process.

We look forward to receiving your revised manuscript.

Kind regards,

Vignesh Muthusamy, PhD

Academic Editor

PLOS One

**Journal Requirements:**

3. Please note that PLOS One has specific guidelines on code sharing for submissions in which author-generated code underpins the findings in the manuscript. In these cases, we expect all author-generated code to be made available without restrictions upon publication of the work. Please review our guidelines at https://journals.plos.org/plosone/s/materials-and-software-sharing#loc-sharing-code and ensure that your code is shared in a way that follows best practice and facilitates reproducibility and reuse.

“The research was supported by the Bill and Melinda Gates Foundation (B&MGF), and the United States Agency for International Development (USAID) through the Stress Tolerant Maize for Africa (STMA, B&MGF Grant # OPP1134248) Project, AGGMW (Accelerating Genetic Gains in Maize and Wheat for Improved Livelihoods, B&MGF Investment ID INV-003439) project and Resilient Maize Hybrids for Sub-Saharan Africa (GF Investment ID INV-088326).”

5. Please note that funding information should not appear in any section or other areas of your manuscript. We will only publish funding information present in the Funding Statement section of the online submission form. Please remove any funding-related text from the manuscript.

6. Please upload a new copy of Figures 3 and 4 as the detail is not clear. Please follow the link for more information:  https://journals.plos.org/plosone/s/figures

Reviewers' comments:

Reviewer's Responses to Questions

**Comments to the Author**

1. Is the manuscript technically sound, and do the data support the conclusions?

Reviewer #1: Yes

Reviewer #2: Yes

2. Has the statistical analysis been performed appropriately and rigorously?

Reviewer #1: Yes

Reviewer #2: Yes

3. Have the authors made all data underlying the findings in their manuscript fully available?

Reviewer #1: Yes

Reviewer #2: Yes

4. Is the manuscript presented in an intelligible fashion and written in standard English?

Reviewer #1: Yes

Reviewer #2: Yes

Reviewer #1: The manuscript presents a comprehensive evaluation of a large number of experimental crosses aimed at developing climate-resilient hybrids, combined with MLN and foliar disease resistance. The findings have significant implications for maize breeding. The following improvements are needed.

Introduction

1.The introduction section is lengthy and difficult to follow.

-Lines 93-94, 96-98, 101-102 and 103-104 are more or less similar to each other.

-Line 103 states the need for DH evaluation, but it is not clear if the lines used were DH. If the lines used were DH, then mention accordingly, else remove this line.

Suggestion: Improve the flow of the introduction section.

Materials and methods

1.Line 19 of the abstract and line 123 mention the used lines as MLN-tolerant, while Table 1 indicates only few lines as MLN-tolerant.

2.Add information on the heterotic grouping of used lines and testers, if available.

3.Screening of TLB and GLS is included in the ‘Assessment of agronomic traits’ subheads. Mentioning them along with MLN screening in a single subhead, such as ‘Screening for resistance to MLN, TLB and GLS,’ would be more appropriate.

4.How reduced senescence was measured is missing in the methodology. Mention at the appropriate place.

5.Line 182 and 185 – it should be jth replicate instead of ith replicate.

6.In statistical analysis – include information on correlation analysis.

Results

1.The result section does not include information on how MLN-tolerant internal checks were used.

Discussion

1.Lines 521-540 contain a repetition of results. It is essential to discuss these findings in the context of their implications and previous results.

Minor comments:

1.Define abbreviations like MCMV and SCMV at their first use in lines 47 and 48.

2.Line 59: the citations should be as [1,5,6,7] instead of [1,5,6][7].

3.Use drought tolerant term uniformly across Table 1. Refer to Lines CKLMARSI0022 and 0029.

Reviewer #2: Comments to the Authors

The manuscript entitled “Genetic Insights from Line × Tester Analysis of Maize Lethal Necrosis Testcrosses for Developing Multi-Stress-Resilient Hybrids in Sub-Saharan Africa” by Gowda et al. is interesting and greatly advances the understanding of maize lethal necrosis for developing multi-stress-resilient hybrids in the era of climate change. The manuscript is presented very well with appropriate data representation through Tables and Figures. The findings are well supported by recent literature in the field with insightful discussion. However, for better clarity and effective conveyance of findings to the scientific community, the following concerns are suggested for the betterment of the manuscript.

Major concerns

Table 2 should be cross-checked for the mean sum of square values, as values are more than σ2G and σ2G×E for most traits. Also need to mention the legend for these terms (σ2G, σ2e, and σ2G×E) in the Table footnotes. It would also be appropriate to mention the degrees of freedom in Table 2.

The CV (%) for ASI under optimum conditions and managed drought is higher than expected, suggesting the need to check for the correctness.

Tables 3, 4, and 5 represent the mean performance of the best 20 testcross hybrids plus seven testers under three different conditions, like MLN disease, optimum, and drought conditions. To facilitate easier comparison of relative performance across conditions, these results could be summarized either using boxplots or consolidated into a single comprehensive table. Accordingly, Tables 3, 4, and 5 may be moved to the Supplementary Material.

Minor concerns:

Some of the keywords could be revised from those other than the words in the manuscript title.

In line 46-51, suggested to cite the more recent literature (2020 onwards) in support of the concerns in maize.

Appropriate legends should be provided in Figures 3 and 4.

Check for appropriate titles and footnotes for all the tables in the manuscript.

**Do you want your identity to be public for this peer review?** For information about this choice, including consent withdrawal, please see our Privacy Policy

Reviewer #1: **Yes:** Vinay Rojaria

Reviewer #2: **Yes:** Ashvinkumar Katral

You may also use PLOS’s free figure tool, NAAS, to help you prepare publication quality figures: https://journals.plos.org/plosone/s/figures#loc-tools-for-figure-preparation

---

## [Author Response · Author response to Decision Letter 1]

5 Feb 2026

PONE-D-25-64892

Genetic Insights from Line × Tester Analysis of Maize Lethal Necrosis Testcrosses for Developing Multi-Stress-Resilient Hybrids in Sub-Saharan Africa

Comments

Editor comments

Response: Thanks for the comment. We formatted the revised manuscript as suggested.

Response: Thanks for the comment. We used field sites managed by CIMMYT and our collaborators. Permits are not required to use field sites, as these experiments are benefiting the partners who own these sites. CIMMYT conduct field trials in these locations every year as part of developing resilient hybrids for the region, which will be provided for them free of cost. The objective of CIMMYT breeding program is to provide new genetics or improved hybrids for the region freely to national or local partners.

3. Please note that PLOS One has specific guidelines on code sharing for submissions in which author-generated code underpins the findings in the manuscript. In these cases, we expect all author-generated code to be made available without restrictions upon publication of the work. Please review our guidelines at https://journals.plos.org/plosone/s/materials-and-software-sharing#loc-sharing-code and ensure that your code is shared in a way that follows best practice and facilitates reproducibility and reuse.

Response: Thanks for the comment. In this experiment the code used for data analyses is from publicly available, standalone software tools like META-R etc. which are freely available for readers and users (data.cimmyt.org/tools).

“The research was supported by the Bill and Melinda Gates Foundation (B&MGF), and the United States Agency for International Development (USAID) through the Stress Tolerant Maize for Africa (STMA, B&MGF Grant # OPP1134248) Project, AGGMW (Accelerating Genetic Gains in Maize and Wheat for Improved Livelihoods, B&MGF Investment ID INV-003439) project and Resilient Maize Hybrids for Sub-Saharan Africa (GF Investment ID INV-088326).”

Response: Thanks for the comment. We included the additional sentence as “The funders had no role in study design, data collection and analysis, decision to publish, or preparation of the manuscript” in the Funding section in the revised manuscript (P27, L729-731).

5. Please note that funding information should not appear in any section or other areas of your manuscript. We will only publish funding information present in the Funding Statement section of the online submission form. Please remove any funding-related text from the manuscript.

Response: Thanks for the comment. Except in Funding section, the funding information did not appear in any section of the manuscript

6. Please upload a new copy of Figures 3 and 4 as the details are not clear. Please follow the link for more information: https://journals.plos.org/plosone/s/figures

Response: Thanks for the comment. We included the improved versions of the figures in the revised manuscript.

Response: Thanks for the comment. We included the captions for all supplementary information in the revised manuscript.

Response: Thanks for the comment, we would recheck if any related comments raised.

Response: Thanks for the comment. We reviewed the reference list and updated it wherever it is needed.

Reviewers' comments:

1. Is the manuscript technically sound, and do the data support the conclusions?

Reviewer #1: Yes

Reviewer #2: Yes

2. Has the statistical analysis been performed appropriately and rigorously?

Reviewer #1: Yes

Reviewer #2: Yes

3. Have the authors made all data underlying the findings in their manuscript fully available?

Reviewer #1: Yes

Reviewer #2: Yes

4. Is the manuscript presented in an intelligible fashion and written in standard English?

Reviewer #1: Yes

Reviewer #2: Yes

Reviewers' comments:

Reviewer #1:

Comment 1

The manuscript presents a comprehensive evaluation of a large number of experimental crosses aimed at developing climate-resilient hybrids, combined with MLN and foliar disease resistance. The findings have significant implications for maize breeding. The following improvements are needed.

Introduction

1.The introduction section is lengthy and difficult to follow.

-Lines 93-94, 96-98, 101-102 and 103-104 are more or less similar to each other.

-Line 103 states the need for DH evaluation, but it is not clear if the lines used were DH. If the lines used were DH, then mention accordingly, else remove this line.

Suggestion: Improve the flow of the introduction section.

Response: Thanks for the comment, we improved the Introduction by removing redundant sentences in the revised manuscript.

Comment 2

Materials and methods

1.Line 19 of the abstract and line 123 mention the used lines as MLN-tolerant, while Table 1 indicates only few lines as MLN-tolerant.

Response: Thanks for the comment. We used elite lines, where most of them are tolerant to MLN, also good for other traits too. So, we modified the relevant sentence in both abstract and materials and methods section as “Thirty-eight early- to intermediate-maturing maize inbred lines, including MLN-tolerant and high-yielding genotypes with drought tolerance and resistance to multiple foliar and insect pests, were crossed with 29 single-cross testers to generate 437 testcross hybrids. These hybrids were evaluated under managed MLN inoculation, drought stress, and optimum conditions across multiple locations.” Please see P7, L144-148.

Comment 3

2.Add information on the heterotic grouping of used lines and testers, if available.

Response: Thanks for the comment. We provided the heterotic grouping information in Table 1.

Comment 4

3.Screening of TLB and GLS is included in the ‘Assessment of agronomic traits’ subheads. Mentioning them along with MLN screening in a single subhead, such as ‘Screening for resistance to MLN, TLB and GLS,’ would be more appropriate.

Response: Thanks for the comment. We separated the disease screening contents in to separate sub-heading in the revised manuscript, P8, L192-208.

Comment 5

4.How reduced senescence was measured is missing in the methodology. Mention at the appropriate place.

Response: Thanks for the comment. We included the relevant information in the revised manuscript, P9, L218-220.

Comment 6

5.Line 182 and 185 – it should be jth replicate instead of ith replicate.

Response: we corrected the mistake in the revised manuscript

Comment 7

6.In statistical analysis – include information on correlation analysis.

Response: we included the relevant information in the revised manuscript, P10, L242-245.

Comment 8

Results

1.The result section does not include information on how MLN-tolerant internal checks were used.

Response: Thanks for the comment. we included the relevant information “Two MLN-resistant internal checks recorded grain yields greater than 3 t ha⁻¹ while maintaining MLN disease severity scores of less than 3.5.” in the revised manuscript, P13, L328-330.

Comment 8

Discussion

1.Lines 521-540 contain a repetition of results. It is essential to discuss these findings in the context of their implications and previous results.

Response: Thanks for the comment. We rewrote the whole section in the revised manuscript, P24, L619-637.

Minor comments:

1.Define abbreviations like MCMV and SCMV at their first use in lines 47 and 48.

Response: Abbreviations are defined

2.Line 59: the citations should be as [1,5,6,7] instead of [1,5,6][7].

Response: corrected in the revised manuscript

3.Use drought tolerant term uniformly across Table 1. Refer to Lines CKLMARSI0022 and 0029.

Response: corrected in the revised manuscript

Reviewer #2: Comments to the Authors

The manuscript entitled “Genetic Insights from Line × Tester Analysis of Maize Lethal Necrosis Testcrosses for Developing Multi-Stress-Resilient Hybrids in Sub-Saharan Africa” by Gowda et al. is interesting and greatly advances the understanding of maize lethal necrosis for developing multi-stress-resilient hybrids in the era of climate change. The manuscript is presented very well with appropriate data representation through Tables and Figures. The findings are well supported by recent literature in the field with insightful discussion. However, for better clarity and effective conveyance of findings to the scientific community, the following concerns are suggested for the betterment of the manuscript.

Response: Thanks for the constructive comments. We addressed all the comments below.

Comment 1

Major concerns

Table 2 should be cross-checked for the mean sum of square values, as values are more than σ2G and σ2G×E for most traits. Also need to mention the legend for these terms (σ2G, σ2e, and σ2G×E) in the Table footnotes. It would also be appropriate to mention the degrees of freedom in Table 2.

Response: Thanks for the comments. Table 2 is variance components estimated by using mixed linear model, whereas Table 3 has mean sum of squares which has DF and other details. In Table 2 we included the foot note as suggested.

Comment 2

The CV (%) for ASI under optimum conditions and managed drought is higher than expected, suggesting the need to check for the correctness.

Response: Thanks for the comments. There was mistake in decimals, we corrected it in the revised manuscript. A high CV for ASI is common in maize breeding trials, particularly under stress conditions. ASI is highly sensitive to drought, heat, and nutrient stress, and strong genotype × environment interactions cause differential responses among lines, increasing variability across plots. In addition, residual heterozygosity or poor flowering synchrony in some DH-derived materials, combined with reduced heritability of ASI under stress, further inflates phenotypic variation.

Comment 3

Tables 3, 4, and 5 represent the mean performance of the best 20 testcross hybrids plus seven testers under three different conditions, like MLN disease, optimum, and drought conditions. To facilitate easier comparison of relative performance across conditions, these results could be summarized either using boxplots or consolidated into a single comprehensive table. Accordingly, Tables 3, 4, and 5 may be moved to the Supplementary Material.

Response: Thanks for the comments. We moved Tables 3, 4 and 5 into supplementary materials and developed boxplots which summarize these results and included in the revised manuscript as Figure 3.

Minor concerns:

Some of the keywords could be revised from those other than the words in the manuscript title.

Response: We revised the key words

In line 46-51, suggested to cite the more recent literature (2020 onwards) in support of the concerns in maize.

Response: We included latest literatures to support our results

Appropriate legends should be provided in Figures 3 and 4.

Response: We modified the Figures 3 and 4 with appropriate legends

Check for appropriate titles and footnotes for all the tables in the manuscript. Response: We modified the Tables with appropriate Titles in the revised manuscript.

---

## [Editor Report · Decision Letter 1]

8 Feb 2026

Genetic Insights from Line × Tester Analysis of Maize Lethal Necrosis Testcrosses for Developing Multi-Stress-Resilient Hybrids in Sub-Saharan Africa

PONE-D-25-64892R1

Dear Dr. Gowda,

We’re pleased to inform you that your manuscript has been judged scientifically suitable for publication and will be formally accepted for publication once it meets all outstanding technical requirements.

Kind regards,

Vignesh Muthusamy, PhD

Academic Editor

PLOS One

Additional Editor Comments (optional):

All the comments of the both the reviewers have been addressed and the improved manuscript is suitable for publication.
---

## [Editor Report · Acceptance letter]

PONE-D-25-64892R1

PLOS One

Dear Dr. Gowda,

I'm pleased to inform you that your manuscript has been deemed suitable for publication in PLOS One. Congratulations! Your manuscript is now being handed over to our production team.

Kind regards,

on behalf of

Dr. Vignesh Muthusamy

Academic Editor

PLOS One